# Impact of stratiform liquid water clouds on vegetation albedo quantified by coupling an atmosphere and a vegetation radiative transfer model

Kevin Wolf[1], Evelyn Jäkel[1], André Ehrlich[1], Michael Schäfer[1], Hannes Feilhauer[2,3,4], Andreas Huth[5,6,2], Alexandra Weigelt[7], and Manfred Wendisch[1]

[1]Leipzig Institute for Meteorology (LIM), Leipzig University, Leipzig, Germany
[2]iDiv German Centre for Integrative Biodiversity Research Halle-Jena-Leipzig, Leipzig, Germany.
[3]Institute for Earth System Science & Remote Sensing, Leipzig University, Leipzig, Germany.
[4]Remote Sensing Centre for Earth System Research, Leipzig University, Leipzig, Germany.
[5]Department of Ecological Modeling, Helmholtz Centre for Environmental Research–UFZ Leipzig, Leipzig, Germany.
[6]Institute for Environmental Systems Research, University of Osnabrück, Osnabrück, Germany.
[7]Systematic Botany and Functional Biodiversity, Institute of Biology, Leipzig University, Germany.

**Correspondence:** Kevin Wolf (kevin.wolf@uni-leipzig.de)

**Abstract.** This paper investigates the influence of clouds on vegetation albedo. For this purpose, we use coupled atmosphere-vegetation radiative transfer (RT) simulations combining the library for Radiative transfer (libRadtran) and the vegetation Soil Canopy Observation of Photosynthesis and Energy fluxes (SCOPE2.0) model. Both models are iteratively linked to more realistically simulate cloud–vegetation-radiation interactions above three types of canopies represented by the spherical, erectophile, and planophile leaf angle distributions. The coupled models are applied to simulate solar, spectral and broadband irradiances under cloud-free and cloudy conditions, with the focus on the visible to near-infrared wavelength range from 0.4 to 2.4 μm wavelengths. The simulated irradiances are used to investigate the spectral and broadband effect of clouds on the vegetation albedo. Changes in solar zenith angle and cloud optical thickness are found to be equally important for variations in vegetation albedo.

The iterative coupling of both models showed that especially the albedo of canopies with an erectophile leaf angle distribution below optically thin clouds in combination with small solar zenith angles is overestimated when a fixed illumination is assumed. For solar zenith angles less than $50°$–$60°$ the vegetation albedo is increased by clouds by up to 0.1. The greatest increase in albedo is observed during the transition from cloud-free to cloudy conditions with a cloud optical thickness ($\tau$) in the range between 0 and 6. For higher values of $\tau$, the albedo of the vegetation saturates and increases only slightly. The increase in vegetation albedo is a result of three effects that are quantified by the simulations: (i) dependence of the canopy reflectivity on the direct and diffuse fraction of downward irradiance, (ii) the shift in the weighting of downward irradiance due to scattering and absorption by clouds, and (iii) multiple scattering between the top of canopy and the cloud base. The observed change in vegetation albedo due to cloudiness is parameterized by a polynomial function, representing a potential method to include cloud–vegetation-radiation interactions in numerical weather prediction and global climate models.

# 1 Introduction

The Earth's surface represents an important boundary between the lithosphere and atmosphere, through which energy fluxes (latent and sensible heat, turbulence, gases, aerosol particles, and radiation) are exchanged. Land-surface–atmosphere interactions are a key concern in dynamic modeling (Ardaneh et al., 2025). In the context of radiative processes, the spectral surface albedo $\alpha(\lambda)$, with $\lambda$ the wavelength, determines the extent to which solar radiation is absorbed and reflected by the Earth's surface. The surface albedo determines the surplus of energy that is transferred into sensible and latent heat (Moene and Van Dam, 2014). Consequently, the integrated surface albedo $\alpha$ is a central factor in numerical weather prediction models and global climate models. Both types of models simulate the interaction between the atmosphere and the surface, and a realistic representation is crucial since cloud–vegetation interactions via surface flues, for example, can reinforce the thickness of shallow stratocumulus (Freedman et al., 2001; Zhang and Klein, 2013). However, the implementation of vegetation albedo in numerical weather prediction and global climate models is often simplified by using climatologies but neglecting cloud–vegetation–radiation interactions (CVRIs) for example in the Integrated Forecasting System (IFS) of the European Centre for Medium-Range Weather Forecasts (ECMWF) (ECMWF, 2021).

In the visible–near-infrared wavelength range (VNIR, 0.3–1.0 µm), bare, dry soils typically have a high albedo, while vegetated surfaces usually exhibit a lower albedo close to zero, particularly for wavelength shorter than 700 nm. This is a result of the large fraction of photosynthetically active radiation (PAR, e.g., Qin et al., 2018) within 0.4 to 0.7 µm wavelength (Roderick et al., 2001; Nemani et al., 2003; Dye, 2004; Min, 2005). In contrast, vegetation has a relatively high reflectivity in the shortwave–infrared (SWIR, 1.0–2.5 µm), compared to bare soil rich in nutrients and moist (Bowker, 1985). In many cases, natural surfaces are a combination of vegetation on bare soil for which the albedo at the top of the canopy (TOC) is often considered the most relevant surface for atmosphere ground interaction.

The TOC albedo is determined by all individual components of the vegetated surface (i.e., leaves, stems, soil and water content) and the structure, for example leaf clumping, of the canopy (Jones and Vaughan, 2010). Most important is the leaf area index (LAI, Watson, 1947; Asner, 1998; Jones and Vaughan, 2010), which can range from 0 to 12. It provides a measure of the total one-sided leaf area per unit of ground area, given in units of $m^2\,m^{-2}$. The LAI itself depends on the vegetation type, follows an annual cycle, and is modulated by climate conditions (Eugster et al., 2000; Davidson and Wang, 2004, 2005). A lower LAI is typically associated with an albedo that approaches the albedo of the soil, whereas an increase in LAI yields an albedo driven by the properties of the canopy (Jones and Vaughan, 2010). The second most important factor controlling canopy radiation characteristics is the leaf angle distribution (LAD, Baldocchi et al., 2002; Jones and Vaughan, 2010; Verrelst et al., 2015; Yang et al., 2023) in combination with the solar zenith angle. The LAD relates the leaf normal and the direction of the incoming radiation, which provides a measure of the sunlit leaf area with respect to the total leaf area. It is therefore a quantitative measure to describe the interaction of incoming radiation with the canopy (Asner, 1998; Stuckens et al., 2009; Vicari et al., 2019). Steeper leaf angles, such as those parameterized by an erectophile LAD, result in lower reflectivity and vice versa (Ollinger, 2011). Therefore, the combination of LAI and LAD determines how the incoming radiation will interact with the canopy and are therefore important parameters that control the CVRI.

In addition to the surface and vegetation characteristics, the TOC albedo is influenced by the atmosphere, clouds that scatter and absorb radiation, and the solar zenith angle. Optically thin clouds are characterized by high transmission with radiation preferably scattered in the forward direction. The remaining fraction is scattered in the backward direction and absorption by water vapor dominates the SWIR part of the solar spectrum. Therefore, incident radiation is attenuated stronger by absorption in the SWIR part, which causes a shift in the relative weighting of the incoming radiation towards shorter wavelengths (Warren, 1982). Furthermore, scattering at cloud particles leads to an increase in below-cloud diffuse radiation. This is particularly relevant for CVRIs, given that diffuse radiation is reflected in no particular direction (isotropic), whereas direct radiation is partly diffused and partly reflected in a preferred direction (specular or Fresnel reflection).

The impact of clouds on CVRIs with regard to snow and ice surfaces were already investigated in Arctic regions by Wiscombe and Warren (1980), Warren (1982), and Stapf et al. (2020). These authors have demonstrated that an increase of the liquid water path (increase in $\tau$) results in an increase in the broadband surface albedo. Although vegetated surfaces have a lower spectral albedo compared to Arctic regions, it can be expected that clouds have a similar effect on the TOC albedo. For example, Gueymard (2017) showed that clouds can enhance the broadband albedo, also called albedo enhancement, by backscattering radiation at cloud base towards to surface, which leads to an increase in the diffuse downward irradiance. Neglecting potential albedo enhancements in models may cause biases in the simulated radiative budget. Furthermore, it is known that very thin cloud layers with $\tau \leq 6$ tend to increase the diffuse downward irradiance that can penetrate deeper into the canopy and enhance the photosynthesis rate, which is also called the diffuse fertilization effect. A further increase in $\tau$ then leads to an overall reduction in downward irradiance and lower photosynthesis rates (Freedman et al., 2001; Min and Wang, 2008; Kanniah et al., 2012). So far, the impact of clouds on TOC albedo and vegetated areas has been neglected in RT simulations. Previous investigations focused on the impact of aerosol and molecular scattering, and on reflectance measurements over vegetation (Ranson et al., 1985; Deering and Eck, 1987; Liu et al., 1994). Some studies exist, for example by Lyapustin and Privette (1999), Myhre et al. (2005), and Yang et al. (2020), who used atmospheric RT simulations to calculate surface reflectances depending on different ratios of downward direct and diffuse radiation. However, fixed ratios of direct and diffuse radiation are assumed in reflectance simulations above vegetation (Atzberger, 2004; Kötz et al., 2004; Schaepman-Strub et al., 2006). Consequently, reflectance simulations, which neglect cloud effects, are spectrally distorted compared to, for example, measurements that are performed under cloudy conditions (Schaepman-Strub et al., 2006; Damm et al., 2015). The spectral distortion is a consequence of cloud–radiation interactions including scattering, transmission, or absorption. The relative contribution of these processes depends on the cloud microphysics, cloud morphology, wavelength of the incident radiation, and canopy structure.

Various sophisticated atmospheric radiative transfer (RT) models include clouds in the simulations. This study will utilize the library for Radiative transfer (libRadtran, Emde et al., 2016). While atmospheric RT models, for example the MODerate resolution atmospheric TRANsmission model (MODTRAN, Berk et al., 2014), have been coupled with vegetation RT models, like the Soil Canopy Observation of Photosynthesis and Energy fluxes version 2 (SCOPE2.0, Yang et al., 2020), none of the previous approaches considered clouds in these simulations. Some studies have either investigated the radiative effects of clouds over different land types and changing forests (Betts, 2000; Bounoua et al., 2002; Cerasoli et al., 2021), while other

studies have concentrated on the RT within or at TOC, taking into account the properties of the canopy itself (Sinoquet et al., 2001; Majasalmi and Rautiainen, 2020; Henniger et al., 2023).

As a result of this discussion and since the radiation interactions of clouds and vegetation have not been explicitly simulated yet, the following four questions are addressed in this paper:

i  How strongly do clouds impact the spectral and broadband albedo of vegetation?

ii  How large are the improvements in broadband albedo by coupling atmospheric and vegetation RT models?

iii  Can we separate and quantify individual coupling effects?

iv  What are the consequences for cloud radiative effects?

To answer the above questions and to systematically investigate CVRIs, we iteratively coupled the atmospheric RT model libRadtran and the vegetation RT model SCOPE2.0 to investigate the radiative interaction of clouds and vegetation. The model coupling provides a more realistic input to the atmospheric radiative transfer model libRadtran by incorporating the vegetation albedo from SCOPE2.0, while the simulated spectral downward irradiance from libRadtran fed into SCOPE2.0 accounts for scattering and absorption by clouds.

The model coupling is introduced in Section 2 by first defining the fundamental properties to describe the RT in the atmosphere and vegetation, and its interaction with the surface. Then the general model set-up is outlined and the basics of the RT models libRadtran and SCOPE2.0 are introduced. The coupling itself is realized by an iterative approach that is applied for different test cases. Section 3 presents the simulations, with Subsection 3.1 outlining the differences between uncoupled and coupled simulations. In Subsection. 3.2 the spectral effects of clouds on the vegetation albedo are shown and in Subsection. 3.3 the impact of clouds on the forest albedo is quantified by running the coupled model for a set of scenarios, including a range of clouds, solar zenith angles, leaf area index, and three different leaf angle distributions. In Subsection. 3.4 the contribution of multiple-scattering and directional effects to the change in the vegetation albedo are separated. A discussion about the implications of a fixed vegetation albedo on the top of canopy radiative budget is given in Subsection. 3.5 and Subsection. 3.6 outlines the limitations of the idealized simulations. The results are summarized in Section 4.

## 2 Terminology, radiative transfer simulations, and iterative coupling

### 2.1 Terminology

We provide the basic radiometric definitions, terminology, and abbreviations that mainly follow Wendisch and Yang (2012), Schaepman-Strub et al. (2006), and Jones and Vaughan (2010) to facilitate the understanding of this paper.

Radiant energy passing through an area element within a certain time interval that originates from a certain solid angle element is defined as the spectral radiance $I(\lambda)$ in units of W m$^{-2}$ nm$^{-1}$ sr$^{-1}$. The spectral irradiance $F(\lambda)$ is defined by the radiant energy passing through an area element within a certain time interval. $F(\lambda)$ is given in units of W m$^{-2}$ nm$^{-1}$ and can

be separated into the upward $F^{\uparrow}(\lambda)$ and downward $F^{\downarrow}(\lambda)$ components. Both are defined with respect to a horizontal surface area from either the lower or upper hemisphere, respectively. $F^{\downarrow}(\lambda)$ is composed of the direct solar irradiance $F_{\text{dir}}^{\downarrow}(\lambda)$, transmitted through the atmosphere without any interaction, and the diffuse irradiance $F_{\text{dif}}^{\downarrow}(\lambda)$, which was at least once scattered by atmospheric constituents, and thus:

$$F^{\downarrow}(\lambda) = F_{\text{dir}}^{\downarrow}(\lambda) + F_{\text{dif}}^{\downarrow}(\lambda). \tag{1}$$

The direct fraction $F_{\text{dir}}^{\downarrow}(\lambda)$ in relation to $F^{\downarrow}(\lambda)$ is quantified by the ratio $f_{\text{dir}}(\lambda)$ defined by:

$$f_{\text{dir}}(\lambda) = F_{\text{dir}}^{\downarrow}(\lambda)/F^{\downarrow}(\lambda) = 1 - F_{\text{dif}}^{\downarrow}(\lambda)/F^{\downarrow}(\lambda). \tag{2}$$

In theory, the ratio $f_{\text{dir}}(\lambda)$ ranges between a value of 0, indicating no direct radiation, and a value of 1, indicating pure direct radiation. However, pure direct radiation is unrealistic under normal atmospheric conditions. The broadband direct fraction $f_{\text{dir}}$ is obtained by:

$$f_{\text{dir}} = \frac{\int_{\lambda_1}^{\lambda_2} F_{\text{dir}}^{\downarrow}(\lambda)\, \mathrm{d}\lambda}{\int_{\lambda_1}^{\lambda_2} F^{\downarrow}(\lambda)\, \mathrm{d}\lambda} \tag{3}$$

Calculating the ratio between $F^{\uparrow}(\lambda)$ and $F^{\downarrow}(\lambda)$ yields the spectral albedo $\alpha(\lambda)$ (unitless) given by:

$$\alpha(\lambda) = \frac{F^{\uparrow}(\lambda)}{F^{\downarrow}(\lambda)}. \tag{4}$$

The spectrally integrated albedo $\alpha$ (unitless) is obtained by weighting the spectral albedo with $F^{\downarrow}(\lambda)$ by:

$$\alpha = \frac{\int_{\lambda_1}^{\lambda_2} \alpha(\lambda) \cdot F^{\downarrow}(\lambda)\, \mathrm{d}\lambda}{\int_{\lambda_1}^{\lambda_2} F^{\downarrow}(\lambda)\, \mathrm{d}\lambda} \tag{5}$$

and integrating over the wavelength range from $\lambda_1$ to $\lambda_2$. To obtain the broadband solar albedo, $\alpha(\lambda)$ is integrated from $\lambda_1 = 0.2$ to $\lambda_2 = 4.5\,\mu\text{m}$, which is equivalent to measurements with broadband albedometers, i.e., a set of upward and downward looking pyranometers. In the following, the integration of $\alpha(\lambda)$ is limited to the wavelength range from 0.4 to 2.4 µm because of model constraints and indicated with $\alpha_{\text{BB}}$. Often, natural surfaces, such as forests, are a combination of vegetation on bare ground for which the albedo at the TOC is often considered to be the most relevant surface for atmosphere–ground-interaction. In this case, the albedo at the TOC is simply referred to as the albedo $\alpha(\lambda)$.

The primary parameter that describes the radiative properties of a canopy is the leaf area index (LAI, Watson, 1947; Asner, 1998; Jones and Vaughan, 2010). It is a measure for the total one-sided area of leaves per unit ground area given in units of $\text{m}^2\,\text{m}^{-2}$, and can range between values from 0 to 12. The LAI depends on vegetation type and is subject to annual and seasonal variations as well as weather and climate conditions (Eugster et al., 2000; Davidson and Wang, 2004, 2005).

The second most important parameter that controls the RT in the canopy is the leaf angle distribution (LAD, Baldocchi et al., 2002; Jones and Vaughan, 2010; Verrelst et al., 2015; Yang et al., 2023). The leaf angle of an individual leaf is defined as the angle between the leaf normal and the zenith. The LAD contains is obtained over all individual leaf angles of a leaf ensemble.

The LAD ultimately determines the sunlit area of a leaf with respect to the one-sided total leaf area and, thus, the area where reflection and absorption occurs (Asner, 1998; Stuckens et al., 2009; Vicari et al., 2019). Goel (1988) proposed six LADs, with three common types: the spherical distribution, where all leave orientations have the same probability; the erectophile distribution, where the majority of leaves have a preferred vertical alignment; and the planophile distribution, where most of the leaves are horizontally aligned. The erectophile and planophile LAD represent two extreme cases among the LADs. Within models, LADs are described by two-parameter beta distributions, trigonometric functions, or ellipsoidal distributions (Goel and Strebel, 1984; Jones and Vaughan, 2010).

The extinction of radiation by scattering and absorption in homogeneous media can be approximated by the turbid medium model (Kubelka, 1931; Kohanovsky, 2009; Jones and Vaughan, 2010). Within the Earth's atmosphere, scattering and absorption by clouds, aerosol particles, and gas molecules is quantified by the optical thickness $\tau(\lambda)$, which depends on the volumetric extinction coefficient $\beta_{\text{ext}}(\lambda)$ (given in units of $\text{m}^{-1}$). Subsequently, the cloud induced optical thickness is simply referred to as $\tau(\lambda)$. In the simplified case of a homogeneous atmosphere, the extinction of direct solar radiation follows the Lambert–Beer–Bougier-law, which can be expressed as:

$$\tau(\lambda, z) = \beta_{\text{ext}}(\lambda) \cdot z, \tag{6}$$

with the path length $z$. The extinction of $I(\lambda)$ at $z$ is then expressed by:

$$\frac{I(\lambda)_z}{I(\lambda)_0} = e^{-b_{\text{ext}}(\lambda) \cdot z} = e^{-\tau(\lambda)}, \tag{7}$$

where $I(\lambda)_0$ is the direct radiance at the top of atmosphere (TOA), $I(\lambda)_z$ is the direct radiance at a certain penetration depth with $z = 0$ at TOA. Monsi (1953) proposed a similar concept to treat the RT in homogeneous vegetation. The attenuation of direct radiance at penetration depth $z$ is then caused by leaves, which are considered as point scatterers (Kubelka, 1931; Jones and Vaughan, 2010). Then, the extinction coefficient $\beta_{\text{ext}}(\lambda)$ in Eq. 6 and Eq. 7 is replaced by $k_{\text{ext}}$, here called the vegetation extinction coefficient. The penetration depth $z$ is replaced by the LAI. A brief overview of $k_{\text{ext}}$ estimates and the attenuation of $I(\lambda)$ within vegetation is provided in in Section D in the Appendix.

## 2.2 Iterative coupling

The albedo of a surface is primarily controlled by the structural parameters of the vegetation, but is also driven by atmospheric factors, namely the direct and diffuse components of the incident radiation $F^{\downarrow}(\lambda)$, and the angle of the incident radiation on the surface. Please note that the incident angle, in the following referred to as $\theta$, is not necessarily equal to the solar zenith angle $\theta_0$. Both angles are approximately equal for cloud-free atmospheres and low aerosol particle concentrations but increasingly deviate for overcast conditions (e.g., see Wiscombe and Warren (1980)). Atmospheric RT models frequently use standard libraries of forest albedo, such as the library of the International Geosphere Biosphere Programme (IGBP; Loveland and Belward, 1997). Conversely, vegetation RT models do simulate the RT in and directly above the canopy but neglect scattering and absorption by clouds in the atmosphere, and assume a fixed ratio of direct to diffuse ratio of $F^{\downarrow}(\lambda)$. By iteratively coupling vegetation and atmospheric RT models, the atmospheric RT model provides more realistic solar spectra of $F^{\downarrow}_{\text{dir}}(\lambda)$ and $F^{\downarrow}_{\text{dif}}(\lambda)$,

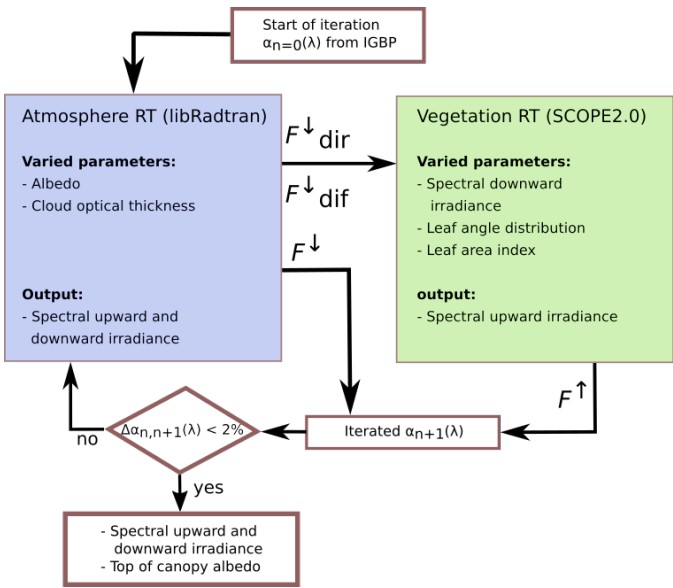

**Figure 1.** Schematic of coupled atmosphere (blue) and vegetation (green) radiative transfer models. The RT models are coupled via the exchange of spectral, direct $F_{\mathrm{dir}}^{\downarrow}(\lambda)$ and diffuse $F_{\mathrm{dif}}^{\downarrow}(\lambda)$ downward irradiance, and the top-of-canopy albedo $\alpha(\lambda)$. The atmospheric RT model is started with a first guess albedo from the IGBP database. When the convergence criteria is met, the iteration is stopped.

while the vegetation RT model provides a more realistic albedo above canopies that is used as input for the atmosphere RT model in the next iteration. In the proposed set-up, the iterative coupling is achieved through the exchange of $F_{\mathrm{dir}}^{\downarrow}(\lambda)$ and $F_{\mathrm{dif}}^{\downarrow}(\lambda)$, and $\alpha(\lambda)$ between the two models. More specifically, $F_{\mathrm{dir}}^{\downarrow}(\lambda)$ and $F_{\mathrm{dif}}^{\downarrow}(\lambda)$ from the atmosphere RT model provide the input to the vegetation RT model. The simulated $F^{\uparrow}(\lambda)$, with $F_{\mathrm{dir}}^{\downarrow}(\lambda)$ and $F_{\mathrm{dif}}^{\downarrow}(\lambda)$ from the atmosphere RT model, allows the calculation of an updated $\alpha(\lambda)$ used as input for the next simulation with the atmosphere RT model.

Figure 1 shows a schematic of the proposed iterative coupling. Each simulation run is realized by $n$ iterations, where each iteration includes a calculation from the atmospheric RT (blue box) and the vegetation RT (green box). The first iteration starts with the atmospheric RT model, using a first guess, spectral surface albedo of forests, here the "mixed forest" spectral surface albedo from the IGBP database. The simulated upward and downward $F_n(\lambda)$ of the first simulation ($n = 0$). The direct and diffuse components of $F_n^{\downarrow}(\lambda)$ are then ingested in the vegetation RT model, which is therefore initialized with $F_n^{\downarrow}(\lambda)$ representing the atmospheric conditions including clouds, instead of a default $F^{\downarrow}(\lambda)$ and direct–diffuse-ratio. The new $F_{n+1}^{\uparrow}(\lambda)$ from the vegetation RT model is then used to calculate $\alpha_{n+1}(\lambda)$ at TOC using Eq. 4. The updated surface albedo provides the input for the atmospheric RT model in the next iteration step. We call an iteration successfully converged if the relative difference between iteration $n$ and $n + 1$ for 90 % of the wavelengths is less than 2 % for the albedo. Formalized, this can be expressed as:

$$P_{90^{th}}\left(\frac{|\alpha_n(\lambda) - \alpha_{n+1}(\lambda)|}{\alpha_{n+1}(\lambda)}\right) < 0.02 \qquad (8)$$

with $P_{90^{th}}$ the $90^{th}$ percentile, the spectral albedo $\alpha_n(\lambda)$ of the previous iteration step, and the spectral albedo $\alpha_{n+1}(\lambda)$ of the current iteration. In this study two iterations were found to be sufficient for all canopy and cloud parameter combinations. This is consistent with Wendisch et al. (2004), who also used an iterative approach to determine the surface albedo from airborne observations. They found that after two iterations, even for rough first estimates of $\alpha(\lambda)$, the retrieved albedo is close to the true surface albedo. In most applications the surface albedo is approximately known, providing a reasonable initial guess, which reduces the number of required iterations.

### 2.2.1 Atmospheric radiative transfer model libRadtran

The atmospheric RT model library for Radiative transfer (libRadtran, Emde et al., 2016) is applied to simulate the RT through the atmosphere above the canopy. The one-dimensional solver "Discrete-Ordinate-Method Radiative Transfer" (DISORT, Stamnes et al., 1988; Buras et al., 2011) is selected to calculate the RT using 12 streams, which is supposed to be sufficient to study irradiances. Since one-dimensional RT models assume homogeneous clouds that best resemble stratiform clouds, this study focuses on low- and mid-level warm stratus and altostratus, which cover between 4 and 12 % of the entire globe at any time (Eastman et al., 2011). These clouds are represented by liquid water clouds with a fixed cloud base at an altitude of 3 km and a cloud top altitude of 3.5 km representative of mid-latitude, continental clouds. The cloud droplet effective radius is fixed to 10 μm that is typically found in such clouds over continents (Stephens, 1994; Frisch et al., 2002; Aebi et al., 2020). The liquid water path is modified such that a desired value of $\tau(\lambda = 550\,\mathrm{nm})$ at 550 nm is achieved and all other values are scaled accordingly, considering the wavelength dependence of $\tau$. The cloud optical thickness $\tau$, is varied between values of 0 and 80 to simulate natural conditions of stratus clouds ranging from cloud-free to densely overcast, and to include values of $\tau$ at which cloud top reflectivity saturates (King, 1987; Nakajima et al., 1991; Tselioudis et al., 1992). Subsequently, $\tau(\lambda = 550\,\mathrm{nm})$ is referred to as $\tau$ for simplicity. Spectral calculations of $F^\uparrow(\lambda)$ and $F^\downarrow(\lambda)$ are performed for a wavelength range from 0.4 to 2.4 μm, which are also used as limits for integrated $F_{\mathrm{BB,sol}}$ in Eq. 3. The incoming spectral irradiance at TOA is represented by the solar reference spectrum provided by Coddington et al. (2021). The solar zenith angle $\theta_0$ is varied between 25° and 70° to cover the typical range of the mid-latitudes. Molecular absorption is considered by using the "medium" resolution parameterization from Gasteiger et al. (2014). A default aerosol distribution after Shettle (1989) is applied, which represents aerosol of rural type in the boundary layer, back-ground aerosol above 2 km during spring-summer, and a visibility of 50 km. Atmospheric profiles of air temperature, humidity, and gas concentrations are represented by the mid-latitude summer profile 'afglms' after Anderson et al. (1986). Absorption by water vapor and other atmospheric trace gases are included in the simulations (Anderson et al., 1986; Emde et al., 2016). The initial run of libRadtran is initialized with the "mixed-forest" albedo $\alpha$ taken from the IGBP data base, which is then replaced by $\alpha(\lambda)$ from SCOPE2.0. An output altitude of 40 m above ground is selected to characterize the downward radiation (direct and diffuse) just above the canopy.

### 2.2.2 Vegetation radiative transfer model SCOPE2.0

The solar RT through vegetation is simulated with the Soil Canopy Observation of Photosynthesis and Energy fluxes (SCOPE2.0 Yang et al., 2021). SCOPE2.0 is an updated version of SCOPE, which has been developed for forward modeling radiances

**Table 1.** Leaf angle distribution (LAD) and corresponding values for the leaf inclination distribution function parameters $\text{LIDF}_a$ and $\text{LIDF}_b$ that were used to parameterize the orientation of the leaves (Yang et al., 2021).

| Distribution | $\text{LIDF}_a$ | $\text{LIDF}_b$ | exemplary specie | | Reference |
|---|---|---|---|---|---|
| spherical | −0.35 | −0.15 | *Tilia cordata* | Small-leaved linden, broadleaf | Pisek et al. (2022) |
| planophile | 1.0 | 0.0 | *Quercus robur* | English oak, broadleaf | Pisek et al. (2022) |
| erectophile | −1.0 | 0.0 | *Ostrya japonica* | Japanese hop-hornbeam, broadleaf | Vicari et al. (2019) |

and albedo for satellite vegetation retrievals. SCOPE2.0 treats the RT by combining the leaf RT model PROperties SPECtra (PROSPECT) with the canopy RT model Scattering by Arbitrarily Inclined Leaves (SAIL Verhoef, 1984; Yang et al., 2017, 2021). At their core, these models are based on the turbid medium approach (Yan et al., 2021). In SCOPE2.0 the soil albedo assumes different soil types, where the moisture dependence is determined by the Brightness-Shape-Moisture (BSM) model (Verhoef et al., 2018; Yang et al., 2020). The default BSM model parameter for soil brightness $B = 0.5$ is used. The optical properties of individual leaves are provided by the Fluorescence spectra (Fluspect) model, which developed out of PROSPECT (Vilfan et al., 2016, 2018). Simulation of vegetation chlorophyll fluorescence was activated in the simulations. The upward directed $F^\uparrow(\lambda)$ and the canopy albedo at TOC is simulated as a superposition of the soil and the contribution from the vegetation. Simulations in the solar part of the spectrum with SCOPE2.0 are generally limited to the wavelength range from 0.4 to 2.4 µm. Consequently, the calculation of spectral $\alpha(\lambda)$ and broadband albedo $\alpha_{\text{BB}}$ are restricted to the same wavelength range, where $\alpha_{\text{BB}}$ is calculated with Eq. 5. The optical properties of vegetation primarily depend on the vegetation type, tree species, tree age, canopy structure, and the solar zenith angle (Liang et al., 2005; Stenberg et al., 2013; Hovi et al., 2017; Zheng et al., 2019). To consider different vegetation states during the annual cycle, the LAI is varied between 1 and 5 $\text{m}^2\,\text{m}^{-2}$, with LAI = 3 $\text{m}^2\,\text{m}^{-2}$ as the default. The lower boundary is selected to include sparse canopies and the upper boundary is selected to include the largest sensitivity of $F^\uparrow(\lambda)$ and $\alpha_{\text{BB}}$ on LAI. For LAI > 5, $F^\uparrow(\lambda)$ and $\alpha_{\text{BB}}$ are expected to become insensitive to changes in LAI, since reflectances start to saturate (Houborg and Boegh, 2008). In SCOPE2.0 the LAD is represented by a linear combination of trigonometric functions in the leaf inclination distribution function (LIDF) (Verhoef, 1998), which is specified by the two parameters $\text{LIDF}_a$ and $\text{LIDF}_b$. Three different LADs - spherical, planophile, and erectophile - are simulated, with parameters $\text{LIDF}_a$ and $\text{LIDF}_b$ provided in Table 1. Table 2 summarizes the relevant parameters for vegetation RT simulations in the visible–near-infrared wavelength range that are kept constant in the simulations.

**Table 2.** Selected configuration of the SCOPE2.0 simulations.

| Description | Symbol | Setting | Unit |
|---|---|---|---|
| leaf chlorophyll concentration | $C_{ab}$ | 40 | µg cm$^{-2}$ |
| leaf carotenoid concentration | $C_{ca}$ | 10 | µg cm$^{-2}$ |
| leaf water equivalent layer | $C_w$ | 0.009 | cm |
| leaf structure parameter | $N$ | 1.5 | Unitless |
| BSM model parameter for soil brightness | $B$ | 0.5 | Unitless |
| volumetric soil moisture content in the root zone | $SMC$ | 0.25 | Unitless |
| vegetation height | $h_c$ | 20 | m |
| output height | $h_{out}$ | 40 | m |

## 3 Results and discussion

### 3.1 Differences between uncoupled and coupled simulations

Analyzing the coupled simulations, it is found that the sensitivity of the simulated spectral and broadband $F(\lambda)$ and $\alpha(\lambda)$ is greatest below $\tau = 6$, thus, defining the range of $\tau$, which is most interesting for understanding CVRIs. Appendix A provides a brief discussion of the response of $\alpha_{BB}$ for $\tau > 6$.

The necessity to initialize vegetation RT simulations with realistic $F^\downarrow(\lambda)$ and the need for RT model coupling is demonstrated in Fig. 2a and b, which show an example of $F^\downarrow_{dif}(\lambda)$ and $\alpha(\lambda)$ at different stages of model coupling and iteration. The

simulations are performed for an intermediate $\theta_0 = 45°$ and $\tau = 2$, where diffuse radiation dominates but direct radiation is still contributing to the radiation field. Under cloud-free conditions (black line), downward diffuse irradiance $F^\downarrow_{dif}(\lambda)$ above the canopy is generally small, except below 700 nm wavelengths, where the contribution from Rayleigh scattering increases. Including clouds in the atmospheric RT model increases $F^\downarrow_{dif}(\lambda)$ (red line) compared to the cloud-free case due to scattering at cloud particles. The spectrum is characterized by water vapor absorption in the wavelength bands of 933–946 nm, 1118–

1144 nm, 1350–1480 nm, and 1810–1959 nm. Coupling of libRadtran and SCOPE2.0 iteratively results in $F^\downarrow_{dif}(\lambda)$ (orange line) which is slightly higher compared to the uncoupled simulations since multiple-scattering enhances $F^\downarrow_{dif}(\lambda)$. Relative differences, given in the subpanel of Fig. 2a, up to $-5$ % are identified between the uncoupled and coupled cloudy simulations for wavelengths between 700 and 1200 nm, where the total $F^\downarrow$ and $\alpha(\lambda)$ are generally largest.

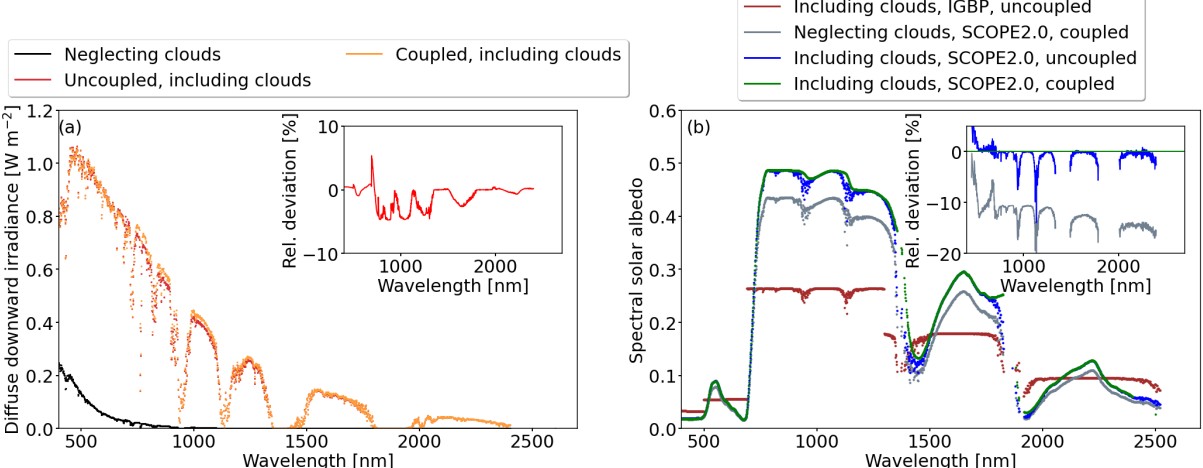

**Figure 2. (a)** Spectral downward diffuse irradiance $F_{\text{dif}}^{\downarrow}(\lambda)$ simulated for cloud-free conditions ($\tau = 0$) and no model coupling (black). $F_{\text{dif}}^{\downarrow}(\lambda)$ simulated for a value of $\tau = 2$ but still in the uncoupled set-up (red). $F_{\text{dif}}^{\downarrow}(\lambda)$ simulated for a value of $\tau = 2$ and with the coupled set-up (orange). **(b)** Spectral albedo $\alpha(\lambda)$ simulated using the "mixed-forest" albedo from the IGBP data base for $\tau = 2$ and uncoupled simulations (brown). $\alpha(\lambda)$ from coupled simulations but neglecting clouds (gray). $\alpha(\lambda)$ from uncoupled simulations including clouds with $\tau = 2$ (blue). $\alpha(\lambda)$ from coupled simulations including clouds with $\tau = 2$ (green). Subplot in **(a)** shows the relative difference between "uncoupled, including clouds" and "coupled, including clouds" with respect to the coupled simulations. Subplot in **(b)** shows the relative difference between "Neglecting clouds, SCOPE2.0, coupled" and "Including clouds, SCOPE2.0, uncoupled" with respect to "Including clouds, SCOPE2.0, coupled".

Since clouds modify $F^{\downarrow}(\lambda)$ spectrally and $f_{\text{dir}}(\lambda)$, they also impact $\alpha(\lambda)$. Figure 2b shows $\alpha(\lambda)$ during different model
set-ups and iterations. A generic $\alpha(\lambda)$ is provided by the IGBP data base (brown line), which was also used as a first-guess to initialize the libRadtran simulations. The radiation is reflected isotropically and does not take into account any dependence on the incident angle nor the presence of clouds. Running SCOPE2.0 freely without any constraints from the atmosphere, i.e., assuming a cloud-free atmosphere, a better resolved $\alpha(\lambda)$ is obtained (gray line). By providing spectra of direct and diffuse $F^{\downarrow}(\lambda)$ that represent cloudy conditions with $\tau = 2$, a higher $\alpha(\lambda)$ is obtained (blue line), which is caused by the greater fraction
of diffuse radiation. Relative differences of about $-2$ to $15$ % are determined. Simulations at this stage of the iteration still neglect CVRIs. Coupling both models under cloud conditions results in $\alpha(\lambda)$ (green line), which is slightly further enhanced compared to the blue line due to multi-scattering between TOC and cloud base. For the given example, the relative differences range between $3$ and $-5$ %, with respect to the fully coupled simulations (see subpanel Fig. 2b). The following analysis will systematically examine the discrepancies in spectral and broadband $F(\lambda)$ and $\alpha$ between uncoupled and coupled simulations,
depending on $\theta_0$ and the optical properties of clouds and vegetation.

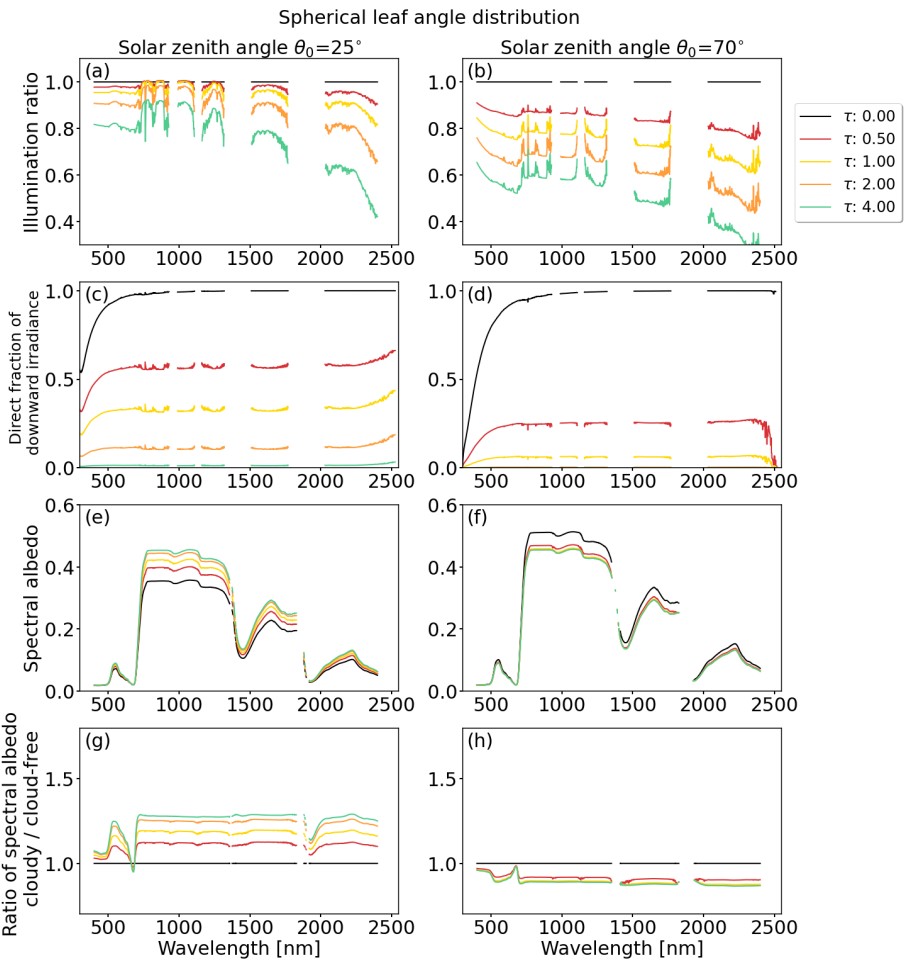

**Figure 3.** Simulations for a solar zenith angle $\theta_0 = 25°$ (left column) and $\theta_0 = 70°$ (right column), a spherical leaf angle distribution, and a leaf area index $\mathrm{LAI} = 3$. Cloud optical thickness $\tau$ is indicated by the colored lines. From top to bottom: **(a,b)** Illumination ratio $F_c^\downarrow(\lambda)/F_{cf}^\downarrow(\lambda)$ (unitless) of spectral downward irradiance $F^\downarrow(\lambda)$ under cloudy conditions (index c) in relation to cloud-free conditions (index cf). **(c,d)** Direct fraction $f_{\mathrm{dir}}(\lambda)$ of total downward irradiance $F^\downarrow(\lambda)$. **(e,f)** Spectral albedo $\alpha(\lambda)$ (unitless). **(g,h)** Illumination ratio $\alpha_c(\lambda)/\alpha_{cf}(\lambda)$ (unitless) of spectral $\alpha$ under cloudy conditions (index c) in relation to cloud-free conditions (index cf).

## 3.2 Sensitivity of spectral surface albedo

Radiation that interacts with clouds is scattered and absorbed. Wavelengths below 900 nm that are outside the absorption bands are primarily affected by scattering from molecules, aerosol, and cloud particles (Mie, 1908), while absorption dominates longer wavelengths. Example simulations of direct and diffuse $F^{\downarrow}(\lambda)$ for four different values of $\tau$ are given in Appendix B. Here we express the wavelength-dependent effects of scattering and absorption on the total $F^{\downarrow}(\lambda)$ by the illumination ratio $F_{\mathrm{c}}^{\downarrow}(\lambda)/F_{\mathrm{cf}}^{\downarrow}(\lambda)$, where $F_{\mathrm{c}}^{\downarrow}(\lambda)$ represents cloudy (index "c") simulations, while $F_{\mathrm{cf}}^{\downarrow}(\lambda)$ represents cloud-free (index "cf") simulations. It is important to note that simulated $F_{\mathrm{cf}}^{\downarrow}(\lambda)$ with $\tau = 0$, simultaneously represent simulations that neglect the presence of clouds in the atmospheric RT.

Figure 3a and b show the illumination ratio for the extreme cases of $\theta_0 = 25°$ and $70°$, respectively. The presence of clouds results in an illumination ratio that is less than 1, since radiation is scattered at the cloud top and absorbed inside the cloud. For the same cloud, a value of $\theta_0 = 70°$ results in a smaller ratio compared to $\theta_0 = 25°$ due to the longer path length through the cloud, which increases extinction. The longer path path length for $\theta_0 = 70°$ also increases the sensitivity of the illumination ratio on $\tau$. The extinction of radiation by absorption at longer wavelengths exceeds the extinction by scattering at shorter wavelengths. In relative terms, the decrease in the radiation above the cloud compared to the radiation below the cloud is more pronounced at longer wavelengths. This results in a spectral slope in the illumination ratio that steepens from shorter to longer wavelengths. The spectral slope becomes more pronounced with increasing $\tau$ and $\theta_0$, indicating a shift in the weighting of the incoming radiation from longer to shorter wavelengths (Wiscombe and Warren, 1980; Grenfell and Perovich, 2008). To illustrate, an increase in $\tau$ from 0 to 1 (yellow line) results in a ratio of 0.95 at 500 nm and a ratio of about 0.9 at 1600 nm. Increasing $\tau$ from 0 to 4 (light green line) results in ratios of 0.75 at 500 nm and 0.65 at 1600 nm wavelengths.

Scattering at clouds changes the fraction $f_{\mathrm{dir}}(\lambda)$ of direct radiation, which determines how radiation is reflected by a surface. Non-isotropic, also called non-Lambertian surfaces, reflect diffuse radiation mostly in a diffuse manner. In contrast, direct radiation reflected by non-isotropic surfaces has a preferred direction that depends on the incident angle and the inherent reflective properties of the surface (Wiscombe and Warren, 1980; Warren, 1982; Grant, 1987; Martonchik et al., 2009). Figure 3c and d show $f_{\mathrm{dir}}(\lambda)$ for $\theta_0 = 25°$ and $\theta_0 = 70°$, respectively. Independent of $\tau$, $f_{\mathrm{dir}}(\lambda)$ is generally low below 700 nm wavelengths as a result of Rayleigh scattering, while $f_{\mathrm{dir}}(\lambda)$ remains relatively constant for wavelengths above 700 nm. The direct fraction $f_{\mathrm{dir}}(\lambda)$ depends on the combination of $\tau$ and $\theta_0$, and is characterized by an increasing sensitivity to larger values of $\theta_0$ due to the longer path lengths of radiation through the cloud.

Figure 3e and f show $\alpha(\lambda)$ for $\theta_0 = 25°$ and $\theta_0 = 70°$, respectively, and a spherical LAD. Figure 3g and h show the related change in $\alpha(\lambda)$ quantified by the ratio $\alpha_{\mathrm{c}}(\lambda)/\alpha_{\mathrm{cf}}(\lambda)$ between cloudy and cloud-free conditions. Please recall, $\alpha_{\mathrm{cf}}(\lambda)$ represent cloud-free conditions and simulations that neglect clouds in the atmospheric RT. The sign and magnitude of the response of $\alpha(\lambda)$ to $\tau$ is controlled by $\theta_0$. For a small value of $\theta_0 = 25°$, the spectral albedo increases compared to the cloud-free simulations, indicated by a ratio $\alpha_{\mathrm{c}}(\lambda)/\alpha_{\mathrm{cf}}(\lambda)$ that is always greater than one and approximately constant over the entire wavelength range.

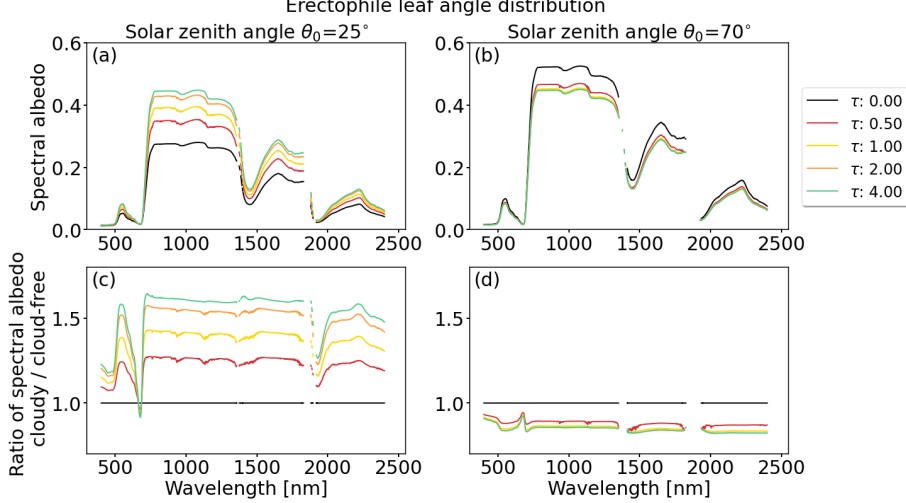

**Figure 4.** Simulations for solar zenith angles of $\theta_0 = 25°$ (left column) and $\theta_0 = 70°$ (right column), an erectophile leaf angle distribution, and a leaf area index LAI = 3. Cloud optical thickness $\tau$ is indicated by the colored lines. From top to bottom: **(a,b)** Spectral albedo $\alpha(\lambda)$ (unitless). **(c,d)** Ratio $\alpha_c(\lambda)/\alpha_{cf}(\lambda)$ (unitless) of spectral $\alpha$ under cloudy conditions (index c) in relation to cloud-free conditions (index cf).

With increasing $\tau$ (decreasing $f_{dir}(\lambda)$) the extinction of $F_{dir}^\downarrow(\lambda)$ and its angular dependence on $\theta_0$ becomes less impor-
tant as isotropic $F_{dif}^\downarrow(\lambda)$ dominates. For the optically thinnest cloud ($\tau = 0.5$) the enhancement is about 10 %. The maximum enhancement for the optically thickest cloud ($\tau = 5$) is between 25 % (864 nm) and up to 40 % (2400 nm) compared to the cloud-free state. For further increasing $\tau$ the change in $\alpha(\lambda)$ becomes smaller and reaches an asymptotic value. For $\theta_0 = 70°$, $f_{dir}(\lambda)$ is generally low even for small values of $\tau$. An increase in $\tau$ from 0 to 0.5 causes a decrease in $\alpha(\lambda)$ of about 10 %, but only marginally increases for a further increase in $\tau$. The decrease is attributed to the lower directional reflectivity of diffuse
radiation compared to direct radiation for same illumination geometry.

Canopies with predominantly vertically oriented leaves are best described by the erectophile LAD. The vertical orientation of the leaves reduces the probability that a photon interacts with the leaves and is scattered out of the canopy (Ollinger, 2011). The lower probability of interaction inside the canopy is formalized in the vegetation extinction coefficient $k_{ext}$, which is lower for the erectophile than for the spherical LAD for $\theta_0$ below 52° (see right column in Table D1 and Eq. 7). In cloud-free
conditions, when $F_{dir}^\downarrow(\lambda)$ dominates, the deeper penetration depth also increases the probability of the radiation being absorbed by the surface. Due to the larger influence of the soil, $\alpha(\lambda)$ for $\theta_0 = 25°$ (Fig. 4a) is generally lower compared to the spherical LAD (Fig. 3e) particularly for the cloud-free case. The narrower erectophile LAD is more sensitive to $\theta_0$ and the transition from direct to diffuse radiation. This leads to a greater variability in $\alpha(\lambda)$ under $\theta_0 = 25°$ compared to the spherical LAD. In cloud-free conditions, $\alpha(\lambda)$ at 850 nm is approximately 0.3 and increased to a maximum of 0.48 for $\tau = 4$. At $\tau = 4$, $\alpha(\lambda)$
approached similar values compared to the spherical LAD. The increase in $\alpha(\lambda)$ from $\tau = 0$ to 4 results in a ratio $\alpha_c(\lambda)/\alpha_{cf}$ of approximately 1.6 except for the absorption bands (Fig. 4c). For $\theta_0$ of 70° (Fig. 4 right column), the response of $\alpha(\lambda)$ on $\tau$, is similar to the behavior found for the spherical LAD. The generally limited response of $\alpha(\lambda)$ on $\tau$ and LAD under large $\theta_0$ is

caused by the dominance of diffuse radiation, where the angular dependent extinction of direct radiation and reflectivity in the canopy becomes negligible.

For the planophile LAD, with mostly horizontally oriented leaves, the area of each leaf and the total probability of interaction with incident radiation is largest compared to the spherical or even the erectophile distribution (Brodersen and Vogelmann, 2007; Gorton et al., 2010). Consequently, $\alpha(\lambda)$ is almost invariant with respect to $\theta_0$ but also $\tau$. For $\tau = 6$, a maximum increase of $\alpha(\lambda)$ by 2 % at 700 nm wavelengths was determined. This is also reflected in an extinction coefficient $k_{\mathrm{ext}}$, which is set to a fixed value of 1, independent of $\theta_0$ (see Table D1 and Fig. D1).

## 3.3  Sensitivity of broadband albedo

### 3.3.1  Impact of cloud optical thickness and solar zenith angle

Figure 5a, d, and g show $\alpha_{\mathrm{BB}}$ as a function of $\tau$ for the spherical, erectophile, and planophile LADs, respectively. Reading Fig. 5a, d, and g along lines of constant $\theta_0$ is interpreted as considering different cloud conditions at a fixed time on any given day. Independent of the LAD and for $\theta_0 \leq 60°$, the broadband $\alpha_{\mathrm{BB}}$ increases with increasing $\tau$. Within one LAD, the increase in $\alpha_{\mathrm{BB}}$ is generally largest for $\theta_0 = 25°$. The sensitivity of $\alpha_{\mathrm{BB}}$ on $\tau$ decreases with increasing $\theta_0$. Comparing the three LADs, the largest variability is found for the erectophile LAD followed by the spherical LAD. For $\theta = 25°$ the transition from cloud-free to overcast conditions ($\tau = 6$) leads to an increase of $\alpha_{\mathrm{BB}}$ by 0.1 for the erectophile LAD and an increase of 0.08 for the spherical LAD. In case of the planophile LAD, $\alpha_{\mathrm{BB}}$ is almost insensitive to $\tau$ with an increase of about 0.002. For $\theta_0 > 60°$, the response of $\alpha_{\mathrm{BB}}$ is reversed for the spherical and the erectophile LAD, where $\alpha_{\mathrm{BB}}$ decreases with increasing $\tau$. Regardless of $\theta_0$ and the LAD, $\alpha_{\mathrm{BB}}$ tends to an asymptotic value of 0.23 when $\tau$ approaches a value of 4, the incoming radiation is dominated by the diffuse component, and $\alpha_{\mathrm{BB}}$ becomes insensitive to changes in $\theta_0$ (e.g., see Fig. 3c,d). Neglecting CVRIs in the simulations, indicated by the dashed lines, results in generally lower values of $\alpha_{\mathrm{BB}}$. The bias is of similar magnitude for all three LADs and increases with increasing $\tau$.

Figure 5b, e, and h show the dependence of $\alpha_{\mathrm{BB}}$ on $\theta_0$ for constant $\tau$. The response of $\alpha_{\mathrm{BB}}$ along the lines of constant $\tau$ represents the diurnal cycle of the Sun under constant cloud conditions. In the case of the spherical and erectophile LAD, an increase in $\theta_0$ is associated with an increase in $\alpha_{\mathrm{BB}}$. The change in $\alpha_{\mathrm{BB}}$ is largest for cloud-free conditions ($\tau = 0$), being most pronounced for the erectophile LAD, and followed by the spherical LAD. This is due to the angular-dependence of scattering in the canopy, which is more pronounced for the erectophile compared to the planophile LAD (see Appendix Fig. D1). For $\tau = 0$ the transition from $\theta_0 = 25°$ to $70°$ leads to an increase in $\alpha_{\mathrm{BB}}$ by 0.12 for the erectophile LAD and an increase of 0.09 for the spherical LAD, which is similar in magnitude compared to the change of $\tau$ for constant $\theta_0 = 25$. For increasing $\tau$, the sensitivity of $\alpha_{\mathrm{BB}}$ to $\theta_0$ is progressively reduced until it becomes insensitive to $\theta_0$ for $\tau = 6$. As for the sensitivity of $\alpha_{\mathrm{BB}}$ to $\tau$, for an overcast sky that is dominated by diffuse radiation, $\alpha_{\mathrm{BB}}$ becomes insensitive to the angular-dependent extinction of the radiation in the canopy, and thus the Sun's diurnal cycle becomes less influential on $\alpha_{\mathrm{BB}}$. In the case of the planophile LAD, $\alpha_{\mathrm{BB}}$ is almost insensitive to $\theta_0$ irrespective of actual value of $\tau$.

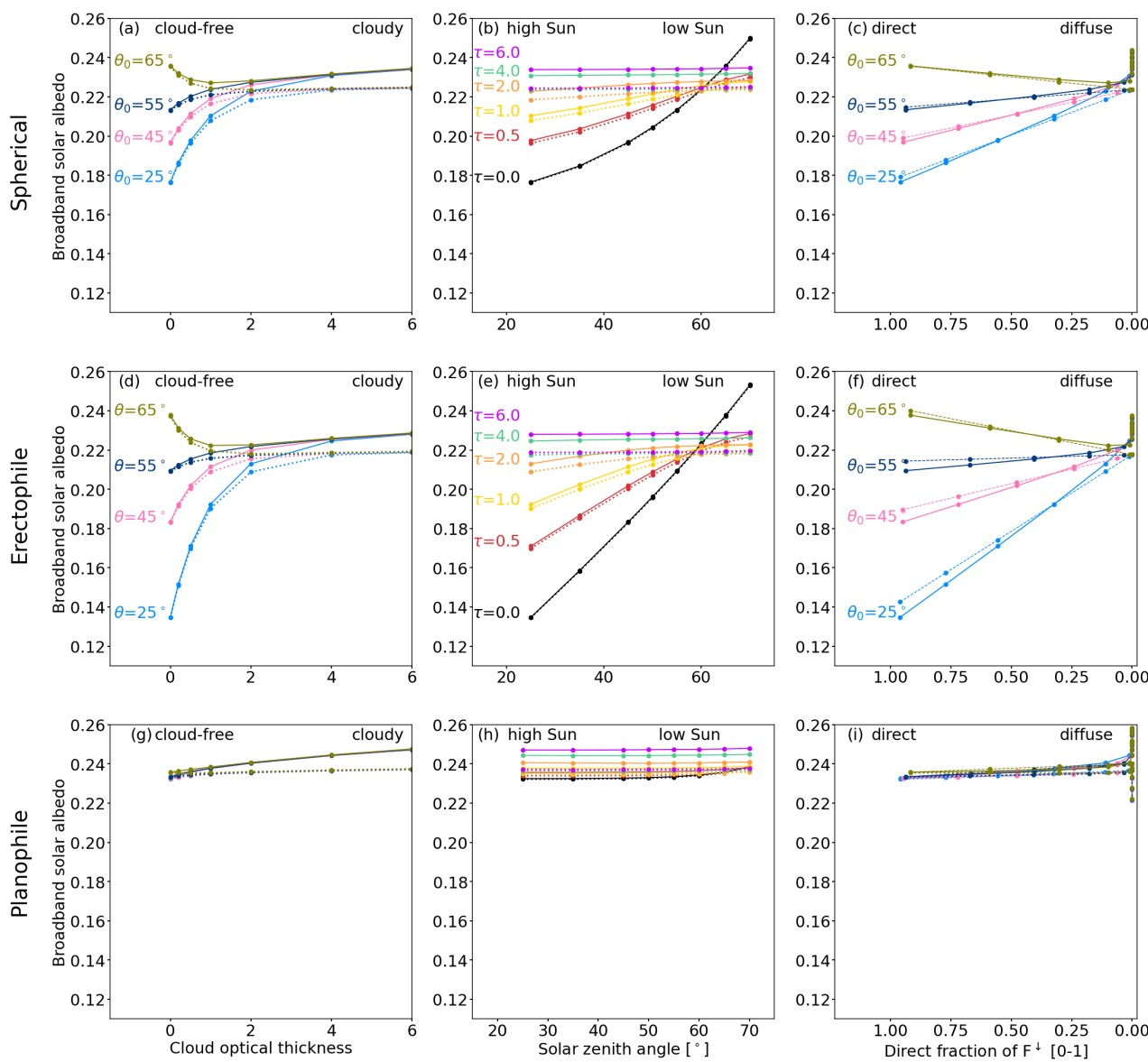

**Figure 5.** First column: Broadband, solar albedo $\alpha_{\mathrm{BB,sol}}$ as a function of cloud optical thickness $\tau$. Second column: $\alpha_{\mathrm{BB,sol}}$ as a function of solar zenith angle $\theta_0$. Third column: $\alpha_{\mathrm{BB,sol}}$ as a function of the direct fraction $f_{\mathrm{dir}}(\lambda)$ of the downward irradiance $F^{\downarrow}$. Lines along $\theta_0$ and $\tau$ are color-coded. Columns from top to bottom provide $\alpha_{\mathrm{BB}}$ based on the spherical, erectophile, and planophile leaf angle distribution, respectively. The dashed lines in the first and second column represent $\alpha_{\mathrm{BB}}$ obtained for uncoupled simulations that neglect cloud–vegetation-radiation interactions. The dashed lines in the third column represent parameterized $\alpha_{\mathrm{BB}}$.

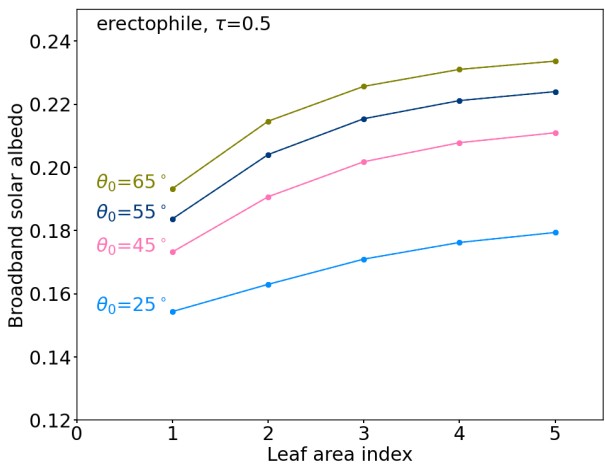

**Figure 6.** Above canopy broadband solar albedo $\alpha_{\mathrm{bb}}$ as a function leaf area index for an erectophile leaf angle distribution and a cloud optical thickness $\tau = 0.5$.

Figure 5c, f, and i show the relationship of $\alpha_{\mathrm{BB}}$ on $f_{\mathrm{dir}}(\lambda)$, which itself depends on $\tau$ and $\theta_0$. Plotting $f_{\mathrm{dir}}(\lambda)$ instead of $\tau$ or $\theta_0$ removes potential ambiguities, since multiple combinations of $\tau$ and $\theta_0$ can lead to the same value of $f_{\mathrm{dir}}(\lambda)$. Furthermore, it removes the exponential relationship in Eq. 7. Moving along lines of constant $\theta_0$ is then synonymous with a change in $\tau$. For the spherical and erectophile LAD, in combination with $\theta_0 < 60°$, $\alpha_{\mathrm{BB}}$ increases with decreasing $f_{\mathrm{dir}}(\lambda)$, while for $60°$ the opposite effect appears. The transition from only direct radiation to only diffuse radiation has the greatest effect for $\theta_0 = 25°$ and decreases with increasing $\theta_0$. For the spherical and erectophile LAD, the lines of constant $\theta_0$ converge and approach an asymptotic value, which indicates that the angular sensitivity of $\alpha_{\mathrm{BB}}$ on $\theta_0$ disappears with increasing cloudiness. The planophile LAD is generally insensitive to changes in $f_{\mathrm{dir}}(\lambda)$ regardless of $\theta_0$.

### 3.3.2 Impact of leaf area index

The LAI is an important parameter that describing the optical properties of a canopy. The simulations in this paper use the SCOPE2.0 default value of 3. Additional simulations with LAIs from 1 to 5 are performed for all three LADs to account for different canopy types, the annual vegetation cycle, and potential leaf loss, for example, due to drought. Figure 6 shows the response of $\alpha_{\mathrm{BB}}$ on LAI under cloudy conditions with $\tau = 0.5$ for the erectophile LAD. Since the LAI describes the leaf area per unit surface area, an increase in the LAI results in a higher probability of incident radiation interacting with the leaves. This is represented in the simulations where $\alpha_{\mathrm{BB}}$ increases with increasing LAI. However, the response of spectral $\alpha(\lambda)$ to changes in LAI is strongly wavelength dependent, and the broadband $\alpha_{\mathrm{BB}}$ is a super-position of two opposing contributions. While for wavelengths greater than 700 nm, an increase in LAI leads to an increase in spectral $\alpha(\lambda)$ because vegetation typically has higher albedo values compared to bare soil in this wavelength range, an increase in LAI results in a decrease in spectral $\alpha(\lambda)$ for shorter wavelengths because in this wavelength range the albedo of vegetation is lower than the albedo of bare, dry soil

(Yang et al., 2021). The response of $\alpha_{\mathrm{BB}}$ to LAI under different $\theta_0$ can be explained by the vegetation extinction coefficient $k_{\mathrm{ext}}(\theta, \lambda)$, which itself depends on wavelength, LAD, and incident angle $\theta$ (Bréda, 2003). The first-order approximation of $k_{\mathrm{ext}}(\theta, \lambda)$ given in Figure D2 and in Appendix D shows that for the same LAD the extinction of radiation depends stronger on LAI when $\theta_0$ is large. This explains the higher sensitivity of $\alpha_{\mathrm{BB}}$ to changes in LAI for larger values of $\theta_0$. Figure D2 also shows that for constant LAI the difference in $k_{\mathrm{ext}}(\theta, \lambda)$, caused by a variation in $\theta_0$, is more pronounced for the erectophile LAD, followed by the spherical and planophile LADs. This explains why the lines of constant $\theta_0$ are well separated for the erectophile LAD shown in Fig. 6, while the lines of constant $\theta_0$ are closer together for the spherical LAD and almost identical for the planophile LAD (both not shown here). Regardless of the LAD, the relationship between LAI and $\alpha_{\mathrm{BB}}$ is generally non-linear. Because of the increasing overlap of leaves with increasing LAI, the increase in additional leaf area does not contribute linearly to the illuminated leaf area that can scatter and absorb incoming radiation.

## 3.4  Separation of coupling effects

### 3.4.1  Contribution of multiple scattering to the enhancement of vegetation albedo

Cloud–vegetation-radiation interactions, here primarily multiple scattering between cloud base and the canopy, are known to enhance the observed spectral albedo (Weihs et al., 2001; Wendisch et al., 2004; Gueymard, 2017). The enhancement is caused by an additional contribution of radiation to $F_{\mathrm{dif}}^{\downarrow}(\lambda)$ that was reflected at the TOC back to the atmosphere and again back to the canopy by the cloud base (Freedman et al., 2001; Min and Wang, 2008; Kanniah et al., 2012; Gueymard, 2017). The relative contribution of CRVIs to the total $F^{\downarrow}(\lambda)$, expressed as $\xi(\lambda)$, is estimated by:

$$\xi(\lambda) = \frac{F_{\mathrm{co}}^{\downarrow}(\lambda) - F_{\mathrm{uco}}^{\downarrow}(\lambda)}{F_{\mathrm{co}}^{\downarrow}(\lambda)}, \tag{9}$$

where $F_{\mathrm{co}}^{\downarrow}(\lambda)$ represents simulated downward irradiance under cloudy conditions from uncoupled (index "uc") simulations and $F_{\mathrm{co}}^{\downarrow}(\lambda)$ represents simulated downward irradiance under the same cloud conditions but from the coupled (index "co") simulations.

Figure 7 shows that $\xi(\lambda)$ is largest between 750 to 900 nm wavelengths, where $\alpha(\lambda)$ and $F^{\downarrow}(\lambda)$ are characterized by their largest values. An exception are the water vapor absorption bands and the red-edge at about 700 nm wavelength. In cloud-free cases ($\tau = 0$, black line), with scattering from molecules and aerosols only, $\xi(\lambda)$ is negligible with a maximum of about $\pm 0.2\,\%$ at 750 nm. With increasing values of $\tau$, which yields more diffuse radiation and a more reflective cloud base, $\xi(\lambda)$ increases continuously at about 600 nm wavelength and for all wavelengths greater than 750 nm. The identified influence of multiple scattering on the surface albedo agrees with earlier observations by Parisi et al. (2003), who identified enhanced diffuse ultra-violet radiation at the surface under cloudy conditions compared to cloud-free conditions. As shown here, similar effects occur also for longer wavelengths in the visible–near infrared spectra particularly where convoluted $\alpha(\lambda)$ with $F^{\downarrow}(\lambda)$ is large. Generally, all cases with high surface albedo, i.e., over snow and ice covered areas, are prone to enhanced diffuse radiation and albedo below clouds (Hay, 1976; Kierkus and Colborne, 1989; Gueymard and Ruiz-Arias, 2016; Gueymard, 2017). Since vegetation albedo is lower than ice and snow covered surfaces, the effects above vegetation are less pronounced.

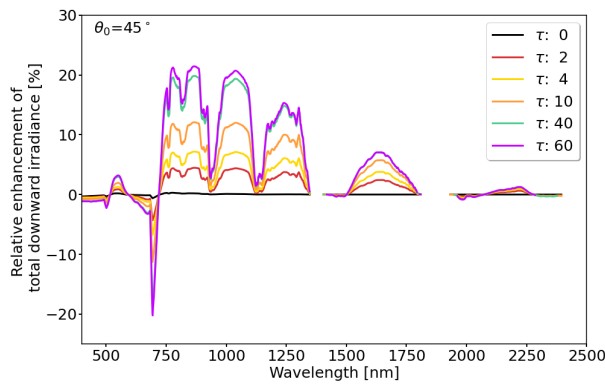

**Figure 7.** Relative contribution $\xi(\lambda)$ (in percent) of downward diffuse irradiance to the enhancement of spectral albedo $\alpha(\lambda)$ due to multiple scattering. An intermediate solar zenith angle $\theta_0$ of $45°$ was selected. Six cloud conditions were considered with cloud optical thickness $\tau$ (unitless) ranging between 0 and 60.

Since $\xi(\lambda)$ represents the relative contribution to $F^{\downarrow}(\lambda)$, the absolute downward $F^{\downarrow}(\lambda)$ is still decreasing with increasing $\tau$. That $\xi(\lambda)$ is driven by the super-position of $\alpha(\lambda)$ and $F^{\downarrow}(\lambda)$ is visible in the spectral slope and the general decrease of $\xi(\lambda)$ with wavelengths.

### 3.4.2 Separating the directional and spectral effects by the downward radiation

In Sec. 3.3 it was shown that $f_{\mathrm{dir}}(\lambda)$ is the main parameter controlling $\alpha_{\mathrm{BB}}$. The individual contributions of direct and diffuse $F^{\downarrow}(\lambda)$ to change in $\alpha_{\mathrm{BB}}$ are quantified by simulating hypothetical cases with either direct or diffuse components of $F^{\downarrow}(\lambda)$. The albedo driven by only direct radiation is commonly referred to as the black-sky albedo, while the albedo that is driven by only diffuse radiation is referred to as the white-sky albedo. Black-sky and white-sky albedo are extreme cases and the actual albedo observed in nature is called blue-sky albedo, which is an intermediate condition between the two extreme cases (Lucht

et al., 2000). Figures 8a–c show $\alpha_{\mathrm{BB}}$ as a function of $\tau$ for the spherical LAD and three values of $\theta_0$ with $25°$, $50°$, and $70°$, respectively. In each panel, the given blue-sky albedo is identical with the graphs given in Fig. 5a. For values of $\theta_0$ of $25°$ and $50°$, $\alpha_{\mathrm{BB}}$ is lowest for the black-sky albedo, while the highest values of $\alpha_{\mathrm{BB}}$ are found for the white-sky albedo. The black-sky and white-sky albedo increase with increasing $\tau$. The blue-sky albedo, as an intermediate condition between the black-sky and white-sky albedo, is closest to the black-sky albedo for cloud-free conditions and approaches the white-sky albedo under

overcast conditions ($\tau > 6$). This agrees with the observations by Freedman et al. (2001) and Kanniah et al. (2012), who found an increase in the canopy albedo under cloudy conditions compared to clear-sky conditions. The different slopes of the blue-sky albedo for different values of $\theta_0$ are caused by the different penetration depth of direct radiation into the canopy. For small values of $\theta_0$ and a dominating direct radiation, the penetration depth into the canopy is high and radiation is more likely to be absorbed, resulting in a lower $\alpha_{\mathrm{BB}}$. Consequently, the difference with respect to the white-sky albedo is greater for $\theta_0 = 25°$

(Fig. 8a) compared to $\theta_0 = 50°$ (Fig. 8b). For even larger values of $\theta_0 = 70°$, increasing $\tau$ results in a decrease of $\alpha_{\mathrm{BB}}$.

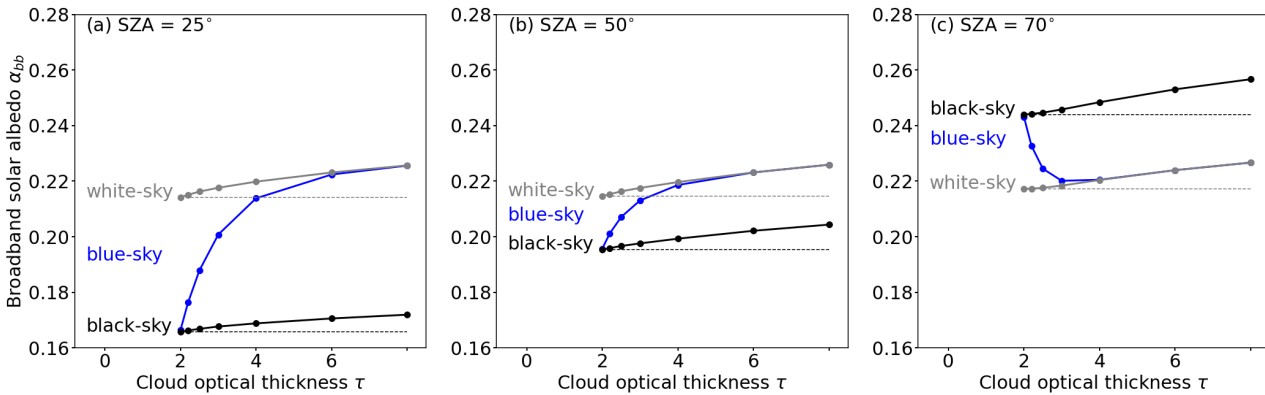

**Figure 8. (a–c)** Broadband, solar albedo $\alpha_{\mathrm{BB}}$ as a function of cloud optical thickness $\tau$ for three solar zenith angles $\theta_0$ of 25, 50, and 70°, respectively. Simulations are performed for a spherical leaf angle distribution and a leaf area index of 3 $\mathrm{m^2\,m^{-2}}$. Simulations including the direct and diffuse fraction of $F^{\downarrow}$ (blue-sky albedo) are given in blue. Simulations including only the direct fraction of $F^{\downarrow}$ (black-sky albedo) are given in black, while broadband albedo including only the diffuse fraction of $F^{\downarrow}$ (white-sky albedo) are given in gray. The dashed lines provide a reference for black-sky and blue-sky albedo.

Broadband $\alpha_{\mathrm{BB}}$ is also modified by spectrally dependent scattering and absorption by clouds that shifts the weighting of $\alpha(\lambda)$ with $F^{\downarrow}(\lambda)$ from longer to shorter wavelengths. These effects are shown for the black-sky and white-sky albedo with respect to the cloud-free state with $\tau = 0$ (dashed lines as reference). For $\theta_0 = 25°$, the black-sky albedo increases by 0.005 and the white-sky albedo increases by 0.06 at $\tau = 8$ compared to the reference at $\tau = 0$. For a value of $\theta_0$ of 70°, the black-sky

albedo increases by 0.06 and the white-sky albedo increases by 0.07 at $\tau = 8$ compared to the reference at $\tau = 0$. Regardless of $\theta_0$, the shift in the weighting of $\alpha(\lambda)$ to shorter wavelengths, enhances the black-sky, white-sky, and blue-sky albedo, but the enhancement is relatively small compared to the overall increase in blue-sky albedo caused by the change in $f_{\mathrm{dir}}$. The effect is small because the weighting is shifted to the wavelength range in which vegetation has the lowest albedo. However, it should be noted that the relative importance of the wavelength shift increases with $\tau$ as the absolute difference between black-sky and

white-sky albedo decreases with increasing $\theta_0$. Furthermore, the effect of the wavelength shift remains effective even when $f_{\mathrm{dir}} = 0$ and diffuse scattering already dominates (see Fig. A in the Appendix).

### 3.5    Consequences for calculating the cloud radiative effect

#### 3.5.1    Effect of neglected cloud–vegetation-radiation interactions

Within ECMWF IFS, the vegetation albedo is based on monthly climatologies of LAI and vegetation type, and thus indirectly

on the LAD (ECMWF, 2021). No separation between black-sky and blue-sky albedo is made (ECMWF, 2021). Consequently, neither the influence of clouds on vegetation albedo nor CVRIs are considered in RT simulations performed within ECMWF IFS or models with similar albedo implementation. Three cases were employed to quantify the differences in the solar radiative energy budget at the top of canopy resulting from neglecting CVRIs and assuming a constant cloud-free vegetation albedo:

- Case A. Neglecting the cloud-induced albedo and CRVIs represents the current albedo implementation of vegetation ECMWF IFS, where the vegetation albedo is set to a constant value. The resulting radiative effect is quantified by the solar radiative forcing $\Delta F$ at the canopy level between the downward broadband irradiance $F^{\downarrow}_{\mathrm{BB,co}}$, obtained from coupled simulations (index "co") including the actual cloud-induced albedo $\alpha_{\mathrm{BB,co}}(\tau)$, and the downward broadband irradiance $F^{\downarrow}_{\mathrm{BB,uc}}$, obtained from uncoupled simulations (index "uc") with a fixed cloud-free albedo $\alpha_{\mathrm{BB,uc}}(\tau=0)$. The solar radiative forcing $\Delta F$ is formalized by:

$$\begin{aligned}
\Delta F = \{ & F^{\downarrow}_{\mathrm{BB,co}}(\alpha_{\mathrm{BB,co}}(\tau),\tau) \\
& -\alpha_{\mathrm{BB,co}}(\tau) \cdot F^{\downarrow}_{\mathrm{BB,co}}(\alpha_{\mathrm{BB,co}}(\tau),\tau)\} \\
& - \{ F^{\downarrow}_{\mathrm{BB,uc}}(\alpha_{\mathrm{BB,uc}}(\tau=0),\tau) \\
& - \alpha_{\mathrm{BB,uc}}(\tau=0) \cdot F^{\downarrow}_{\mathrm{BB,uc}}(\alpha_{\mathrm{BB,uc}}(\tau=0),\tau)\}.
\end{aligned} \quad (10)$$

- Case B. Neglecting CRVIs but accounting for the cloud-induced albedo causes a radiative effect that is quantified by the solar CRVI-forcing $\Delta F_{\mathrm{crvi}}$, calculated at the canopy level between $F^{\downarrow}_{\mathrm{BB,co}}$ from coupled simulations and $F^{\downarrow}_{\mathrm{BB,uc}}$ from uncoupled simulations, both accounting for the actual cloud-induced albedo. The CRVI-forcing $\Delta F_{\mathrm{crvi}}$ is formalized by:

$$\begin{aligned}
\Delta F_{\mathrm{crvi}} = \{ & F^{\downarrow}_{\mathrm{BB,co}}(\alpha_{\mathrm{BB,co}}(\tau),\tau) \\
& -\alpha_{\mathrm{BB,co}}(\tau) \cdot F^{\downarrow}_{\mathrm{BB,co}}(\alpha_{\mathrm{BB,co}}(\tau),\tau)\} \\
& - \{ F^{\downarrow}_{\mathrm{BB,uc}}(\alpha_{\mathrm{BB,uc}}(\tau),\tau) \\
& - \alpha_{\mathrm{BB,uc}}(\tau) \cdot F^{\downarrow}_{\mathrm{BB,uc}}(\alpha_{\mathrm{BB}}(\tau),\tau)\}.
\end{aligned} \quad (11)$$

- Case C. Neglecting the cloud-induced albedo but including CRVIs introduces a bias, which we call the solar albedo forcing $\Delta F_{\mathrm{alb}}$. It is quantified at the canopy level between $F^{\downarrow}_{\mathrm{BB,uc}}$, combined with the cloud-induced albedo $\alpha_{\mathrm{BB,uc}}(\tau)$, and $F^{\downarrow}_{\mathrm{BB,uc}}$, combined with a fixed cloud-free albedo $\alpha_{\mathrm{BB}}(\tau=0)$. The albedo-forcing $\Delta F_{\mathrm{alb}}$ is estimated from uncoupled simulations, since coupled simulations would include CRVI effects. The solar albedo-forcing $\Delta F_{\mathrm{alb}}$ is formalized by:

$$\begin{aligned}
\Delta F_{\mathrm{alb}} = \{ & F^{\downarrow}_{\mathrm{BB,uc}}(\alpha_{\mathrm{BB,uc}}(\tau),\tau) \\
& -\alpha_{\mathrm{BB,uc}}(\tau) \cdot F^{\downarrow}_{\mathrm{BB,uc}}(\alpha_{\mathrm{BB,uc}}(\tau),\tau)\} \\
& - \{ F^{\downarrow}_{\mathrm{BB,uc}}(\alpha_{\mathrm{BB,uc}}(\tau=0),\tau) \\
& - \alpha_{\mathrm{BB,uc}}(\tau=0) \cdot F^{\downarrow}_{\mathrm{BB,uc}}(\alpha_{\mathrm{BB,uc}}(\tau=0),\tau)\}.
\end{aligned} \quad (12)$$

Figure 9a shows $\Delta F$, $\Delta F_{\mathrm{alb}}$, and $\Delta F_{\mathrm{crvi}}$ for a spherical LAD. For $\theta_0$ less than 60°, $\Delta F_{\mathrm{alb}}$ increases with increasing $\tau$, reaches a maximum, and then decreases for further increases in $\tau$. For $\theta_0 < 60°$, the cloud-induced $\alpha_{\mathrm{BB}}$ is greater than the $\alpha_{\mathrm{BB}}$ under cloud free conditions, causing the first term in Eq. 12 to be greater than the second term, resulting in negative $\Delta F_{\mathrm{alb}}$. A peak value of $\Delta F_{\mathrm{alb}} \approx -35\,\mathrm{W\,m^{-2}}$ occurred for the combination of $\theta_0 = 25°$ and $\tau = 4$. For values of $\theta_0$ of 50° and

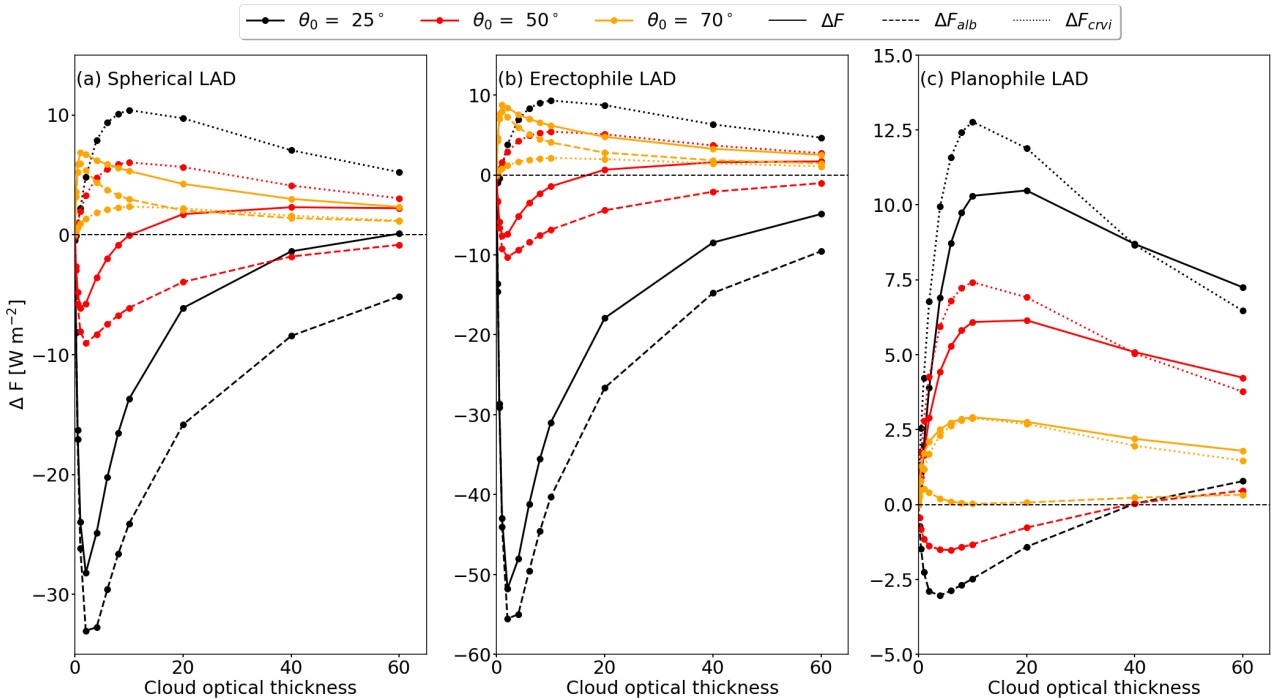

**Figure 9.** Absolute difference in above canopy broadband solar albedo-forcing $\Delta F_{\mathrm{alb}}$ (dashed line), broadband solar CRVI-forcing $\Delta F_{\mathrm{crvi}}$ (dotted line), and resulting broadband solar forcing $\Delta F$ (solid line) due to the cloud-modulated canopy albedo and multiple-scattering effects. Results are given for the spherical **(a)**, erectophile **(b)**, and planophile **(c)** leaf angle distribution, and three solar zenith angle $\theta_0$ of $25°$, $50°$, and $70°$.

70°, peak values of $-8\,\mathrm{W\,m^{-2}}$ and $6\,\mathrm{W\,m^{-2}}$ are determined, respectively. Positive values of $\Delta F_{\mathrm{alb}}$ result from the decrease of $\alpha_{\mathrm{BB}}$ when transitioning from clear-sky to cloud-induced $\alpha_{\mathrm{BB}}$ (see Fig. 5a). Independently of $\theta_0$, further increasing $\tau$ beyond the respective peak values of $\Delta F_{\mathrm{alb}}$ leads to a decrease in $\Delta F_{\mathrm{alb}}$, which is caused by a decrease in $F^{\downarrow}_{\mathrm{BB}}$ that counterbalances the effect of the cloud-induced albedo.

Independent of the Sun's position, the contribution of the CVRIs leads to positive $\Delta F_{\mathrm{crvi}}$, reaching peak values up to $10\,\mathrm{W\,m^{-2}}$ for the combination of $\tau = 6$ and $\theta_0 = 25°$. The CRVI-forcing is positive because $F^{\downarrow}_{\mathrm{BB}}$ and the cloud-induced $\alpha_{\mathrm{BB}}$ are larger for the coupled simulations (first term in Eq. 11) than for the uncoupled simulations (second term in Eq. 11). Like for $\Delta F_{\mathrm{crvi}}$, a further increase of $\tau$ beyond the maxima of $\Delta F_{\mathrm{crvi}}$, leads to a decrease due to the decrease in $F^{\downarrow}_{\mathrm{BB}}$ with $\tau$.

The greatest forcing is related to $\Delta F$, which can be partly understood as a super-position of $\Delta F_{\mathrm{alb}}$ and $\Delta F_{\mathrm{crvi}}$. The solar forcing can be positive or negative depending on the combination of $\theta_0$ and $\tau$. It is noted that $\Delta F_{\mathrm{alb}} + \Delta F_{\mathrm{crvi}} \neq \Delta F$, since different stages of the coupling are used in the calculation of $\Delta F_{\mathrm{alb}}$ and $\Delta F_{\mathrm{crvi}}$. For small values of $\theta_0 = 25°$ and optically thin clouds, $\Delta F$ is negative with values up to $-28\,\mathrm{W\,m^{-2}}$ at $\tau = 6$, becomes smaller with increasing $\tau$, and turns it's sign for optically thick clouds at $\tau = 60$ due to the dominance of $\Delta F_{\mathrm{crvi}}$. For greater $\theta_0$ of about $50°$, $\Delta F$ approaches a peak value of $-7\,\mathrm{W\,m^{-2}}$ at $\tau = 6$, and becomes positive for $\tau \approx 15$. For $\theta_0 = 70°$, $\Delta F$ is positive for all values of $\tau$.

For the erectophile LAD, a similar behavior of $\Delta F$ to $\tau$ is observed, but with greater magnitude in $\Delta F_{\mathrm{alb}}$ and $\Delta F$, resulting in peak $\Delta F$ of $-52\,\mathrm{W\,m^{-2}}$ for $\theta = 25°$ and $10\,\mathrm{W\,m^{-2}}$ for $\theta = 70°$. Compared to a canopy with a spherical LAD, an erectophile canopy generally has a lower reflectivity, resulting in reduced multiple-scattering between TOC and the cloud base. This leads to lower $\Delta F_{\mathrm{crvi}}$ for all values of $\theta_0$.

The planophile LAD, with preferably horizontally oriented leaves, reflects a larger fraction of the incoming radiation that can contribute to the enhancement of $F^{\downarrow}$ below clouds, resulting in the largest values of $\Delta F_{\mathrm{crvi}}$ among all three LADs (Brodersen and Vogelmann, 2007; Gorton et al., 2010). The albedo forcing associated with the planophile LAD was found to be the smallest, not exceeding $-2.6\,\mathrm{W\,m^{-2}}$ for $\theta_0 = 25°$, since $\alpha_{\mathrm{BB}}$ is almost insensitive to changes in $\tau$ (see Fig. 5g–i). Overall, $\Delta F$ is dominated by positive $\Delta F_{\mathrm{crvi}}$, resulting in positive $\Delta F$ for all simulated conditions of $\theta_0$ and $\tau$.

Regardless of the solar zenith angle, $\Delta F_{\mathrm{alb}}$, $\Delta F_{\mathrm{crvi}}$, and $\Delta F$ are most sensitive and reach peak values in the simulated cases when $\tau$ is less than 20. This results from the sensitivity of $\alpha_{\mathrm{BB}}$ and $f_{\mathrm{dir}}$ to $\tau < 6$ and the non-linear behavior of $F^{\downarrow}_{\mathrm{diff}}$, which has a maximum at about $\tau = 4 - 6$ below liquid water clouds (Bohren, 1987). Therefore, the transition from cloud-free to cloudy conditions with $\tau < 20$ is most susceptible to biases, when neglecting the diffuse vegetation albedo and CRVIs.

Neglecting CVRIs and the influence of clouds on the vegetation albedo introduces biases in the surface radiation budget. As shown in Fig. 9, neglecting CVRIs underestimates the amount of diffuse radiation between canopy and cloud base under illumination conditions with $\theta_0 < 60°$. This is explained by the relatively small extinction coefficient for $\theta_0 < 60°$, allowing direct radiation to penetrate deep into the canopy and to be absorbed by the canopy or soil, where it is converted into latent and sensible heat. Both fluxes are known to be important for boundary layer processes and local cloud formation (Freedman et al., 2001; Bosman et al., 2019). In contrast, a generally higher $f_{\mathrm{dir}}$ below clouds and CVRIs increase the probability that radiation is reflected in the upper parts of the canopy and lowering the probability of absorption by the understory or soil. A second potential consequence of neglecting CVRIs is an incorrect estimate of the radiation available for photosynthesis. Due to the presence of clouds, the radiation reflected at the top of the canopy is again reflected at the cloud base that is available as diffuse radiation. It is known that diffuse radiation below optically thin clouds with $\tau \le 6$ increases the amount of photosynthetically active radiation, since diffuse radiation can penetrate deeper into the canopy and all parts of the leaves can absorb radiation, not only the leaf areas illuminated by direct radiation (Freedman et al., 2001; Min and Wang, 2008; Kanniah et al., 2012; Cheng et al., 2016). In the same manner, the enhancement of diffuse radiation by CRVI potentially increases photosynthesis rates that will be underestimated otherwise. This results in a potential underestimation of plant productivity and carbon uptake (Freedman et al., 2001).

### 3.5.2 Parameterization of the cloud effect on broadband surface albedo

To better approximate the effect of clouds on the vegetation albedo, we propose a parameterization of $\alpha_{\mathrm{BB}}$ as a function of broadband $f_{\mathrm{dir}}$ to account for CVRIs and the cloud induced vegetation albedo (Fig. 5). The parameterization takes atmospheric parameters $\theta_0$ and $f_{\mathrm{dir}}$, and the vegetation parameters LAI and LAD as input. The parameterization of $\alpha_{\mathrm{BB}}$ is formalized by:

$$\alpha_{\mathrm{BB}} = g(\mu) \cdot f_{\mathrm{dir}} + b_1 \cdot LAI^2 + b_2 \cdot LAI + b_3 \tag{13}$$

**Table 3.** Parameters and polynomials for the parameterized broadband solar albedo $\alpha_{\mathrm{BB}}$. Maximal deviations $\Delta\alpha_{\mathrm{BB}}$ between simulation and parameterization.

| Leaf angle distribution | $a_1$ | $a_2$ | $a_3$ | $a_4$ | $b_1$ | $b_2$ | $b_3$ | $\Delta\alpha_{\mathrm{BB}}$ |
|---|---|---|---|---|---|---|---|---|
| spherical | −0.0490 | 0.1722 | −0.2839 | 0.1059 | −0.0038 | 0.0346 | 0.1618 | 0.003 |
| erectophile | −0.2310 | 0.3587 | −0.3694 | 0.1340 | −0.0021 | 0.0216 | 0.1670 | 0.008 |
| planophile | −0.0633 | 0.1483 | −0.1166 | 0.0229 | −0.0037 | 0.0330 | 0.1747 | 0.002 |

where $\mu = \cos(\theta_0)$ and $g(\mu)$ is given by:

$$g(\mu) = a_1 \cdot \mu^3 + a_2 \cdot \mu^2 + a_3 \cdot \mu + a_4. \tag{14}$$

The parameters $a_1$ to $a_4$ and $b_1$ to $b_3$ for the spherical, erectophile, and planophile LAD are provided in Table 3.

The parameterization of $\alpha_{\mathrm{BB}}$ is evaluated against the simulated values of $\alpha_{\mathrm{BB}}$ and is overlaid in the right column of Fig. 5. The values of $\alpha_{\mathrm{BB}}$ from the simulations and the parameterization mostly overlap, indicating a good agreement of the parameterization with the simulations. Regardless of the LAD, discrepancies appear mainly when $f_{\mathrm{dir}}$ approaches a value of 0. General largest differences appear for the erectophile LAD, but remain below a value of $\Delta\alpha_{\mathrm{BB}} = 0.005$, which corresponds to a relative error of 2.3% with respect to $\alpha_{\mathrm{BB}} = 0.22$. Since the proposed parameterization takes $f_{\mathrm{dir}}$ as input, the parameteri-

zation only accounts for the transition from direct to diffuse radiation, i.e., the transition from cloud-free to the cloud-induced $\alpha_{\mathrm{BB}}$. The shift in the spectral weighting, which persists even when $f_{\mathrm{dir}} = 0$, is not considered. However, the contribution of the wavelength shift is generally small compared to the effect of $f_{\mathrm{dir}}$ as shown in Fig. 8 and Appendix A.

### 3.6 Limitations of the simulations

libRadtran and SCOPE2.0 allow to specify a variety of parameters during the simulation set-up. While certain parameters such

as $\theta_0$, $\tau$, LAI, or LAD were varied in the present study, other parameters were left at their respective model defaults from libRadtran (Section 2.2.1) and SCOPE2.0 (Section 2.2.2). Since this idealized set-up does not cover the natural variability of atmospheric and vegetation conditions, the chosen default values may affect the results presented in this study. An additional sensitivity analysis of selected default parameters, which potentially impact the RT in the solar wavelength range, is performed. Details about the sensitivity analysis, the varied parameters, and the value ranges are provided in Appendix Section C It is noted

that the sensitivity study does not cover all possible parameters in both models and is therefore not comprehensive. It should be regarded as a first-order approximation of potential uncertainties associated with the fixed parameters and provide an estimated for the robustness of the presented results.

Within the varied atmospheric parameters, the variation of the vertical temperature and relative humidity profile showed the largest impact on $\alpha_{\mathrm{BB}}$ with an increase of up to +0.01, when the mid-latitude summer profile ("afglms", default) was

exchanged with the mid-latitude winter profile ("afglmw" Anderson et al. (1986)). Effects of aerosol concentration, cloud altitude, and cloud droplet size were found to be of minor importance with a variation in $\alpha_{\mathrm{BB}}$ below $\pm 0.008$. Among the

varied vegetation parameters, the largest effect on $\alpha_{\mathrm{BB}}$ was found for plant dry matter. Varying the plant dry matter by $\pm 25\%$ around its default value of $0.0012\,\mathrm{g\,cm^{-2}}$ resulted in a variation in $\alpha_{\mathrm{BB}}$ of $\pm 0.013$, which is of similar magnitude compared to a change in the LAI from 2 to 3. The second most influential factor in the sensitivity analysis was the leaf structure parameter, which is known to be an uncertainty factor in vegetation RT and modeling (Boren et al., 2019). However, the variation of $\alpha_{\mathrm{BB}}$ due to changes in the leaf structure parameter is smaller than the effects reported from changes in $\tau$ and $\theta_0$. Variation of in chlorophyll AB content, carotenoid content, leaf water equivalent layer, model parameter for soil brightness, volumetric soil moisture content in the root zone resulted in absolute deviations in $\alpha_{\mathrm{BB}}$ of $\pm 0.003$ maximum. Neither the present study nor the sensitivity analysis considered the influence of canopy structure. Canopy structure is known to be a key factor in determining the amount of radiation that is absorbed, reflected, and transmitted in a canopy (Ni-Meister et al., 2010). For example, clumping reduces the sunlit area of the leave ensemble compared to randomly oriented leaves for the same LAI. This affects the interaction of incoming radiation and consequently the canopy albedo (Ni and Woodcock, 2000; Chen et al., 2008). All simulations performed are based on the assumption of a homogeneous canopy to cover a wider range of canopy types, since clumping depends on vegetation type, among other factors. By not accounting for leaf clumping, the amount of radiation absorbed by the sunlit leaf area overestimated (Li et al., 2023) and neglecting clumping in our simulations may reduce the dependence on $\theta_0$ and $\tau$ in our simulations (Kanniah et al., 2012; Li et al., 2023).

## 4    Summary and conclusions

This study investigated cloud–vegetation-radiation interactions (CVRIs) by coupling an atmospheric radiative transfer (RT) model, the library for radiative transfer (libRadtran), and a vegetation RT model, the Soil Canopy Observation of Photosynthesis and Energy fluxes (SCOPE2.0). This goes beyond previous model set-ups, where vegetation RT models neglected the influence of clouds, which are now explicitly included in the coupled radiative transfer simulations.

The coupled simulations were run for an interval of solar zenith angles $\theta_0$ ranging from $25°$ to $70°$. A stratiform liquid water cloud was simulated with cloud optical thickness $\tau$ ranging from 0, for cloud-free conditions, to 80, for fully overcast conditions. The range of $\tau$ is intended to represent a typical mid-latitude spring, summer, or autumn day. The diversity of plant characteristics was attempted to be partly represented by spherical, erectophile, and planophile leaf angle distributions (LADs), and variations of the leaf area index (LAI) between 1 and 5 $\mathrm{m^2\,m^{-2}}$ (inclusive). The simulations by libRadtran and SCOPE2.0 covered a wavelength range from 0.4 to 2.4 μm. The iterative coupling was realized by initializing SCOPE2.0 with the spectral, downward direct $F_{\mathrm{dir}}^{\downarrow}(\lambda)$ and diffuse irradiance $F_{\mathrm{dif}}^{\downarrow}(\lambda)$ provided by libRadtran. libRadtran was initialized with a first guess vegetation albedo, which was replaced in the next iteration step with the vegetation albedo provided by SCOPE2.0. Two cycles were found to be sufficient for the iteration to converge.

The iterative coupling allowed to account for the change in the direct fraction under cloudy conditions and CRVIs in the calculation of the cloud-induced vegetation albedo. An example case showed that initializing SCOPE2.0 with direct and diffuse downward irradiance under cloudy conditions enhanced the spectral vegetation albedo $\alpha(\lambda)$ by about 10 to 15 % compared to cloud-free conditions. The inclusion of CRVIs resulted in a further increase of about 1 to 5 %. The enhancement was found to

be wavelength dependent with the largest relative differences near the water vapor absorption bands and where high values of $\alpha(\lambda)$ and total downward irradiance $F^{\downarrow}(\lambda)$ coincide.

    Based on the varied parameters and parameter ranges, it was found that the LAD is the primary factor controlling the sensitivity of $\alpha_{\mathrm{BB}}$ to LAI, $\theta_0$, and $\tau$. Assuming an erectophile LAD in the simulations, $\alpha_{\mathrm{BB}}$ was most sensitive to the varied parameters, especially for combinations of small $\tau < 6$ and small $\theta_0 < 50°$, i.e., large values of the direct fraction $f_{\mathrm{dir}}(\lambda)$.

Generally lower sensitivities of spectral and broadband $\alpha$ to $\tau$ and $\theta_0$ were found for the spherical LAD. Spectral and broadband $\alpha$ of the planophile LAD were found to be almost insensitive to $\tau$ and $\theta_0$ for the same parameter ranges. The sensitivity of $\alpha(\lambda)$ to LAI, LAD, and $\theta_0$ decreased continuously with decreasing fraction $f_{\mathrm{dir}}(\lambda)$ because the incident radiation becomes more diffuse, i.e., undirected, and the angular-dependent scattering in the canopy becomes insensitive to the canopy structure given by LAI and LAD. The second effect that affected the spectrally integrated broadband albedo $\alpha_{\mathrm{BB}}$ was the wavelength-

dependent absorption and scattering by clouds, which shifted the weight of the incoming radiation towards shorter wavelengths. Due to the generally low values of $\alpha(\lambda)$ below 700 nm, the effect of the wavelength shift was found to be small in absolute values, increasing $\alpha_{\mathrm{BB}}$ by up to 0.07 ($\theta_0 = 70°$ and $\tau = 6$). In summary, the change in $f_{\mathrm{dir}}(\lambda)$ was found to be relevant for values of $\tau$ between 0 and 6, when direction radiation is dominant. Beyond $\tau$ of 6, the shift in the spectral weighting of $\alpha(\lambda)$ with $F^{\downarrow}(\lambda)$ was found to be the main contributor to changes in $\alpha_{\mathrm{BB}}$.

Different stages of the iterative process were used to separate the effects of diffuse radiation on $\alpha(\lambda)$ from the effects of multiple scattering. Iterative coupling was found to be particularly important to account for multiple scattering between the top of the canopy and the cloud base, which enhanced $F^{\downarrow}_{\mathrm{dif}}(\lambda)$ by up to 22 % between 750 and 900 nm wavelengths for a cloud with cloud optical thickness $\tau = 60$.

    The radiative effect of clouds on $\alpha_{\mathrm{BB}}$ and the resulting radiation budget below clouds was estimated in terms of the solar

forcing $\Delta F$ at the top of the canopy. The solar forcing $\Delta F$ was determined between uncoupled simulations neglecting the influence of clouds on vegetation albedo and coupled simulations including the cloud effects on vegetation albedo. The solar forcing was further decomposed into the solar albedo-forcing $\Delta F_{\mathrm{alb}}$, representing the bias due to a fixed vegetation albedo, and the solar CRVI-forcing $\Delta F_{\mathrm{crvi}}$, representing the bias by missing CRVI. The greatest sensitivity of $\Delta F$ was found for the transition from cloud-free to cloudy conditions ($\tau < 6$). The largest absolute values of $\Delta F$ were identified for $\theta_0 = 25°$,

leading to negative $\Delta F$ of up to $-58\,\mathrm{W\,m^{-2}}$, implying a stronger reflection by vegetation in the coupled simulations compared to uncoupled simulations that neglected the influence of clouds. The maximum values of $\Delta F$ decreased with increasing $\theta_0$ and also reversed sign, so that for $\theta_0 = 70°$, $\Delta F$ became positive, with values up to $8\,\mathrm{W\,m^{-2}}$. The contributions of $\Delta F_{\mathrm{alb}}$ and $\Delta F_{\mathrm{crvi}}$ to $\Delta F$ were found to depend on the combination of LAD, $\theta_0$, and $\tau$ since both components can have opposite signs. For the spherical and erectophile LAD, $\Delta F_{\mathrm{alb}}$ dominated in most cases, while for the planophile LAD, $\Delta F_{\mathrm{crvi}}$ dominated $\Delta F$.

The nearly linear correlation between $\alpha_{\mathrm{BB}}$ and $f_{\mathrm{dir}}$ has been exploited to parameterize the effect of clouds on $\alpha_{\mathrm{BB}}$ over vegetated areas. The parameterization accounts for $\theta$, LAI, LAD, and $f_{\mathrm{dir}}$. It has been shown that the parameterization is able to reproduce the simulated cloud-induced albedo changes with a relative error of less than 2.4 %. The approach to parameterize the effect of clouds on $\alpha_{\mathrm{BB}}$ over vegetated areas may be suitable for implementation in numerical weather prediction or global circulation models to improve the surface radiation budget over vegetated areas under cloudy conditions.

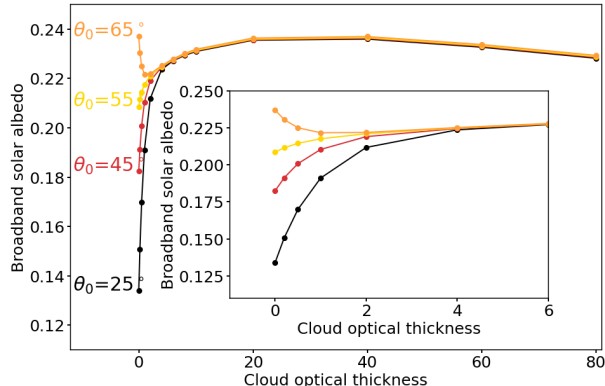

**Figure A1.** Above canopy broadband solar albedo $\alpha_{\mathrm{bb}}$ as a function of cloud optical thickness $\tau$ ranging from 0 to 80 and for four solar zenith angles, with a default leaf area index of 3. An erectophile leaf angle distribution is assumed.

The idealized simulation set-up and the multitude of vegetation parameters did not allow to cover the natural variability of atmospheric and vegetation conditions, and the chosen default values may affect the results presented in this study. A sensitivity study was performed to estimate the influence of these default parameters and to test the robustness of the results. Among the varied parameters, plant dry matter had the largest effect on $\alpha_{\mathrm{BB}}$, followed by the assumed atmospheric profile and the leaf structure parameter. Varying the default values by $\pm 25\,\%$ caused deviations in $\alpha_{\mathrm{BB}}$ of up to $\pm 0.013$, which corresponds to a change in LAI of about 1. Variations in aerosol visibility, cloud altitude, and cloud droplet effective radius contributed only little. Variations in chlorophyll AB content, carotenoid content, leaf water equivalent layer, BSM model parameter for soil brightness, and the volumetric soil moisture content in the root zone had only small effects. It is further acknowledged that the simulations assume a homogeneous canopy and structural effects such as leaf clumping were not considered in this study. However, these structural effects operate on a local scale and are likely to be smoothed out given the current spatial resolution of numerical weather prediction models and global circulation models.

## Appendix A:  Sensitivity of broadband solar albedo on the full range of cloud optical thickness

Coupled simulations of spectral irradiance $F(\lambda)$ and albedo $\alpha(\lambda)$ have been performed for cloud optical thickness $\tau$ with values between 0 and 80. Integration of $\alpha(\lambda)$ using Eq. 5 results in the broadband $\alpha_{\mathrm{BB}}$ weighted by the incoming $F^{\downarrow}(\lambda)$. Spectral dependent scattering and absorption by clouds shifts the relative weighting towards shorter wavelengths. Figure A1 shows the response of $\alpha_{\mathrm{BB}}$ on $\tau$ for the erectophile leaf angle distribution (LAD). Initially, $\alpha_{\mathrm{BB}}$ increases or decreases with increasing $\tau$ until the diffuse component of $F^{\downarrow}(\lambda)$ dominates at $\tau = 6$. This increase is related to the transition from only direct ($\tau = 0$) to diffuse ($\tau = 6$) downward irradiance $F^{\downarrow}(\lambda)$. Beyond a value of $\tau = 6$, the further increase of $\alpha_{\mathrm{BB}}$ is only related to the shift of the weighting in $F^{\downarrow}(\lambda)$ to shorter wavelengths. The spectral slope of the incoming radiation - roughly decreasing with increasing wavelength - and the spectral slope of the vegetation - low $\alpha(\lambda)$ below 700 nm, steep increase,

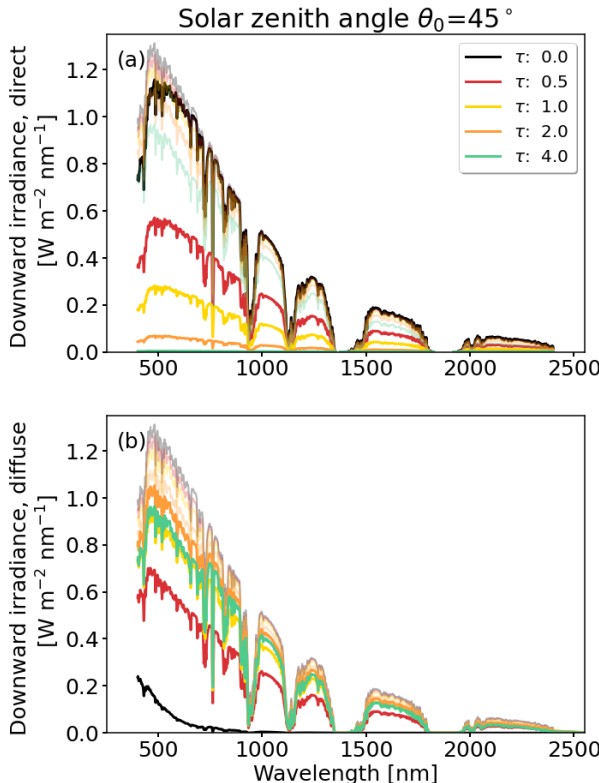

**Figure B1.** Panel **(a)** and **(b)** show spectral, downward, direct $F_{\mathrm{dir}}^{\downarrow}(\lambda)$ and diffuse $F_{\mathrm{dif}}^{\downarrow}(\lambda)$ irradiance, respectively. In both panels spectral, downward, total irradiance $F^{\downarrow}(\lambda)$ is underlaid by faded lines. Cloud optical thickness $\tau$ is indicated by the colored lines. Simulations based on a spherical leaf angle distribution for a solar zenith angle of $\theta_0 = 45°$.

and decreasing with increasing wavelength - lead to a maxima in the convolution of $\alpha(\lambda)$ and $F^{\downarrow}(\lambda)$, such that $\alpha_{\mathrm{BB}}$ becomes maximal at $\tau \approx 20$. Beyond this optimum, $\alpha_{\mathrm{BB}}$ decreases because the spectral weighting in $F^{\downarrow}(\lambda)$ is shifted more and more into the spectral range where the radiation is almost completely absorbed by vegetation. The simulation with the erectophile LAD represents an extreme case. For the spherical and the planophile LAD, a reduced sensitivity of $\alpha_{\mathrm{BB}}$ to $\tau$ between 0 and 6 was found. However, the position of the maximum at around $\tau = 20$ showed to be insensitive to the selected LAD.

**Appendix B: Influence of clouds on downward irradiance**

Radiation passing through the atmosphere is scattered and absorbed by aerosol particles, gas molecules, and clouds. The influence of clouds on the direct irradiance $F_{\mathrm{dir}}^{\downarrow}(\lambda)$ and the diffuse irradiance $F_{\mathrm{dif}}^{\downarrow}(\lambda)$ components of the total irradiance $F^{\downarrow}(\lambda)$ is shown in Fig. B1 for an intermediate solar zenith angle $\theta_0$ of 45°. All spectra are characterized by water-vapor absorption bands at 933–946 nm, 1118–1144 nm, 1350–1480 nm, and 1810–1959 nm wavelengths due to molecular absorption. An

640 increase in $\tau$ results in a decrease in $F_{\mathrm{dir}}^{\downarrow}(\lambda)$ (Fig. B1a). Wavelengths below 900 nm that are outside of the absorption bands

are primarily affected by Rayleigh and Mie scattering (Mie, 1908), leading to a flattening of the spectrum below 500 nm. Wavelengths above 900 nm and within the water–vapor-absorption bands are dominated by absorption. It is further noted that with decreasing / increasing $\theta_0$ the path of the radiation thought the atmosphere and the cloud becomes shorter / longer, leading to less / more scattering processes. Consequently, the same values of cloud optical thickness $\tau$ yield $F_{\mathrm{dir}}^{\downarrow}(\lambda)$ that are greater / lesser for $\theta_0$ lesser / greater than $\theta_0 = 45°$. Radiation scattered at least once by atmospheric constituents is removed from the direct component $F_{\mathrm{dir}}^{\downarrow}(\lambda)$ and contributes to the diffuse component $F_{\mathrm{dif}}^{\downarrow}(\lambda)$ given in Fig. B1b. For the cloud-free case (black), $F_{\mathrm{dif}}^{\downarrow}(\lambda)$ is close to zero except for wavelengths $\lambda < 750$ nm due to Rayleigh scattering. Regardless of $\theta_0$, including clouds in the simulations leads to an overall increase in $F_{\mathrm{dif}}^{\downarrow}(\lambda)$. However, the increase is not continuous and reaches maximum values for $\tau$ between 2 and 4 at $\theta_0 = 25°$ and $\tau$ around 1 at $\theta_0 = 75°$. This is a result of the pronounced forward peak in the scattering phase function of water droplets, which enhances scattering toward the surface compared to cloud-free conditions. According to Bohren (1987), the maximum $F_{\mathrm{dif}}^{\downarrow}(\lambda)$ occurs under cloudy conditions when $\tau \approx \ln(2/(1-g)) \cdot \cos(\theta_0) \approx 2.6$, with $g = 0.85$ the asymmetry factor with a representative value for clouds in the visible-near infrared wavelength range (Irvine and Pollack, 1968).

## Appendix C:  Uncertainty estimates due to selection of default parameters

The robustness and potential uncertainties of the broadband surface albedo $\alpha_{\mathrm{BB}}$ to variations of the input parameters that were previously fixed in the study, an additional sensitivity study was performed. All simulated combinations base on an intermediate solar zenith angle $\theta_0 = 45°$ and a cloud optical thickness $\tau = 4$, which were chosen to represented mean illumination conditions that were targeted in the present paper. Table C1 lists all parameters that were kept previously constant but were varied in the sensitivity study. Table C1 also lists the default values and the absolute values that resulted from a variation by $\pm 25\%$ from their default value. While this may not cover the full range of possible parameters nor represents the full natural variability, the sensitivity study can be regarded as a first-order approximation to estimate the effect of deviating from the default values. A variation of the leaf area index was included to provide a reference between the simulations in the main study and the variational analysis presented here. Further analysis of the effect of vegetation parameters on the canopy RT can be found in Yang et al. (2020) and Yang et al. (2021).

The influence of the atmospheric parameters aerosol visibility (AV), cloud altitude (CA), and cloud droplet effective radius (ER) were found to be small with an influence on $\alpha_{\mathrm{BB}}$ that is below $\pm 0.0005$. For the atmospheric profile (AP) a larger impact on $\alpha_{\mathrm{BB}}$ was determined with 0.007 and 0.011 for the US-standard atmosphere and for the mid-latitude winter atmosphere, respectively (Anderson et al., 1986). Variations in the vegetation parameter also remained small for chlorophyll AB content (Cab), carotenoid content (Cca), leaf water equivalent layer (Cw) with absolute deviations in $\alpha_{\mathrm{BB}}$ with respect to the reference not exciting $\pm 0.003$, and BSM model parameter for soil brightness (BSM), and the volumetric soil moisture content in the root zone (SMC). In relation to that, a variation of the leaf are index (LAI) by $\pm 1$ around the default value caused a deviation in $\alpha_{\mathrm{BB}}$ of $\pm 0.01$. The greatest influence was found for the dry matter content (Cdm) with a deviation of $\pm 0.015$.

**Table C1.** List of parameters varied in the atmospheric radiative transfer simulations to estimate the uncertainty in the broadband surface albedo with respect to a given parameter. Relative differences in the broadband surface albedo of a given parameter are given with respect to the default configuration and related maximal absolute deviations in broadband albedo $\alpha_{\mathrm{BB}}$.

| Parameter | Abbreviation | Unit | Default | Variation | Max. abs. uncertainty in $\alpha_{\mathrm{BB}}$ |
|---|---|---|---|---|---|
| Atmospheric profile | AP | | mid-latitude summer | mid-latitude winter, US-standard profile | 0.0067 |
| Aerosol visibility | AV | km | 50 | 20, 80 | 0.0005 |
| Cloud altitude | CA | km | 3–3.5 | 2.5–3, 3.5–4 | −0.0002 |
| Cloud droplet effective radius | ER | $\mu$m | 10 | 7, 13 | −0.0004 |
| Chlorophyll AB content | Cab | $\mu\mathrm{g\,cm}^{-2}$ | 40.0 | 30, 50 | 0.0035 |
| Carotenoid content | Cca | $\mu\mathrm{g\,cm}^{-2}$ | 10 | 7.5, 12.5 | 0.0003 |
| Dry matter content | Cdm | $\mathrm{g\,cm}^{-2}$ | 0.012 | 0.0, 0.015 | 0.0146 |
| Leaf water equivalent layer | Cw | cm | 0.009 | 0.00675, 0.01125 | 0.0028 |
| Leaf structure parameter | N | | 1.4 | 1.05, 1.75 | −0.0044 |
| BSM model parameter for soil brightness | BSM | | 0.5 | 0.375, 0.625 | −0.0022 |
| Volumetric soil moisture content in the root zone | SMC | | 25 | 18.75, 31.25 | 0.0006 |
| Leaf area index | LAI | $\mathrm{m}^2\,\mathrm{m}^{-2}$ | 3 | 2, 4 | −0.0119 |

**Table D1.** Vegetation extinction coefficients $k_{\text{ext}}(\theta)$ for the spherical, planophile, and erectophile leaf angle distribution taken from Jones and Vaughan (2010).

| Distribution | Approximation of $k_{\text{ext}}(\theta)$ |
| --- | --- |
| spherical | $k = 1/(2 \cdot \cos\theta)$ |
| erectophile | $k = (2 \cdot \tan\theta)/\pi$ |
| planophile | $k = 1$ |

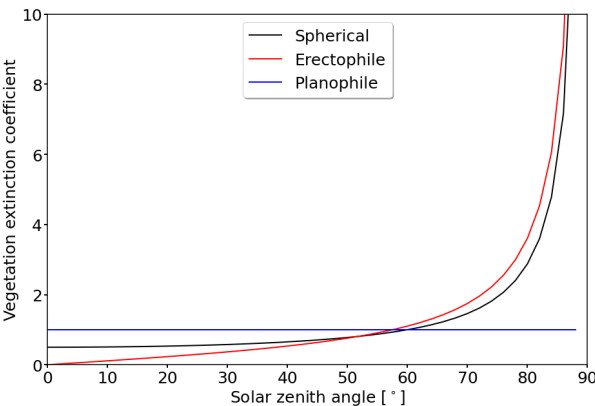

**Figure D1.** Extinction coefficient in dependence of incident angle $\theta$ for the spherical, erectophile, and planophile leaf angle distribution.

## Appendix D: Approximate direct beam extinction in vegetation

Within a homogeneous vegetation layer, the radiative transfer can be approximated by the turbid medium approach (Jones and Vaughan, 2010). The attenuation of direct radiance $I_0(\lambda)$ at the penetration depth $z$ can be expressed by the Equation 7. Among other factors, the vegetation extinction coefficient $k_{\text{ext}}(\theta, \lambda)$ depends on the stand structure and canopy architecture, wavelength, direct and diffuse fraction of incident radiation, and the incident angle $\theta$ (Bréda, 2003). It is therefore not straight forward to determine analytical expressions for $k_{\text{ext}}(\theta, \lambda)$ (Bréda, 2003; Jones and Vaughan, 2010). First order approximations are provided, which do neglect the wavelength-dependence of $k_{\text{ext}}(\theta, \lambda)$. It is also assumed that the solar zenith angle $\theta_0$ is equal to the incident angle $\theta$. However, state of the art vegetation radiative transfer (RT) models such as SCOPE2.0 account for wavelength-dependent effects by using numerical procedures (Yang et al., 2021). In the literature various values of $k_{\text{ext}}(\theta)$ exist, ranging from fixed values (Pierce and Running, 1988; Wan et al., 2021); over empirical, tabulated values (Bréda, 2003); to trigonometric functions that account for the dependence on the incident angle of radiation (Jones and Vaughan, 2010). Figure D1 shows $k_{\text{ext}}(\theta)$ as a function of $\theta$ for the spherical, erectophile, and planophile LAD. The planophile leaf angle distribution (LAD) is approximated with a value of $k_{\text{ext}}(\theta) = 1$. The spherical and erectophile LAD are described by the trigonometric functions given in Table D1. For $\theta < 52°$, $k_{\text{ext}}(\theta)$ of the spherical LAD exceeds $k_{\text{ext}}(\theta)$ of the erectophile LAD. The erectophile LAD is characterized by a steeper slope and, therefore, $k_{\text{ext}}(\theta)$ of the erectophile LAD is more sensitive

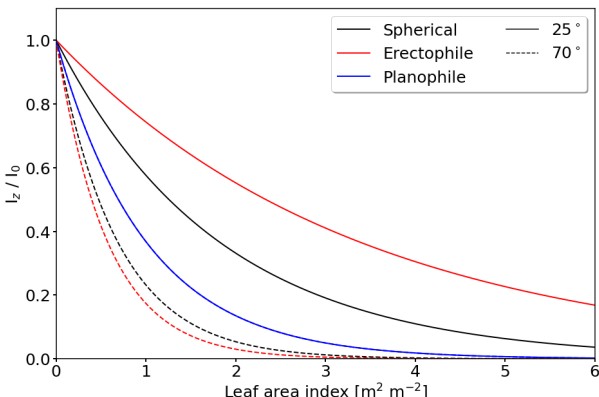

**Figure D2.** Ratio $I_z/I_0$ of direct radiance $I_z$ at penetration depth $z$=LAI calculated with Eq. 7 and direct beam radiance $I_0$ at top of canopy as a function of leaf area index LAI. Two incident angles $\theta$ of 25° and 70° are given.

to changes on $\theta$. For $\theta > 52°$, $k_{\text{ext}}(\theta)$ of the erectophile LAD exceed the spherical LAD resulting in a larger $k_{\text{ext}}(\theta)$ with increasing $\theta$. Note that extinction includes the processes of scattering and absorption, which means that an increase in $k_{\text{ext}}(\theta)$

means an increase in absorption in the canopy, but can also be caused by an increase in scattering.

The estimated values of $k_{\text{ext}}(\theta)$ are used to estimate the extinction of direct radiance in dependence of the LAI. Figure D2 shows that for the Sun near the zenith ($\theta = 25°$) the slope is steepest for the planophile LAD, followed by the spherical and erectophile LAD. The incident direct radiation is reduced to 50 % ($I_z/I_0 = 0.5$), when LAI values of 0.7, 2.3, and 1.26 for the planophile, spherical, and erectophile LADs are exceeded, respectively. For the Sun near the horizon ($\theta = 70°$) the slope

is steepest for the erectophile LAD, followed by the spherical and planophile LAD. The ratio $I_z/I_0 = 0.5$ is reached at LAI of 0.7, 0.4, and 0.5 for the planophile, erectophile, and septically LAD, respectively. As a result, for the default LAI of 3 and $\theta_0 = 70°$ the direct radiation cannot penetrate deep into the canopy, while for same LAI and $\theta_0 = 25°$ the direct radiation can penetrate deepest into the canopy for the erectophile, followed by the spherical and the planophile LAD.

*Author contributions.* **KW** designed and implemented the model coupling, performed the simulations, and drafted the manuscript. **EJ**, **AE**,

**MS**, and **MW** contributed equally to the preparation of the manuscript. **AHu**, **HF**, and **AW** helped with the model set up and the revision of the manuscript.

*Competing interests.* The authors declare no competing interest.

*Acknowledgements.* We thank the German Centre for Integrative Biodiversity Research (iDiv) Halle-Jena-Leipzig, which is a research centre of the Deutsche Forschungsgemeinschaft (DFG). We also thank the Saxon State Ministry for Science, Culture and Tourism (SMWK) for funding thought grant 3-7304/44/4-2023/8846.

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
