# Peer review of "Impact of stratiform liquid water clouds on vegetation albedo quantified by coupling an atmosphere and a vegetation radiative transfer model"

_EGUsphere, 2024_

## Author Comment (AC1)

**Reply to Reviewer #1**
(Referee comment on "Impact of clouds on vegetation albedo quantified by coupling an atmosphere and a vegetation radiative transfer model" by K. Wolf et al. (egusphere-2024-3614),https://doi.org/10.5194/egusphere-2024-3614-RC1, 2025)

We thank the Reviewer for the time she/he spent on the manuscript. We appreciate the provided comments and we believe that the manuscript has been significantly improved by incorporating the Reviewer's suggestions.
To improve readability and logical structure, the order of sections and subsections has been revised. We hope that this will allow the reader to better navigate through the different points discussed in the manuscript. The manuscript has also been extended to include a discussion of modeling limitations and the influence of the selected default values.

For better legibility, the Reviewer's comments are highlighted in **bold** and changes in the manuscript are in *italic*.
* * *
**This manuscript investigates an important component of the Earth system for understanding radiative budgets at the land surface, which has significance for land-atmosphere coupling (carbon, water, energy) --- the influence of an interactive atmosphere and land surface radiative transfer scheme. It is limited to vegetated canopies with certain assumed properties but the authors investigate a range of idealized scenarios to quantify the effect of these interactions. Overall, the manuscript is well-written and structured. It was relatively easy to follow what the authors did and what they found, which is challenging given the complexity and technical nature of the topic. I have provided a few general comments that I believe the authors need to consider before it is suitable for publication and a number of minor comments too. It was probably on the border of minor and major revisions - I opted for major as it should give the authors ample time to address the comments.**

**General Comments**

**The introduction is written very well and the knowledge gap is identified clearly. The only recommendation I have is to add a little bit more introduction (background/discussion) around how vegetation structure has been studied in the past in the context of surface albedo, and by what mechanism it may impact surface albedo. You present two questions toward the end of the Introduction. Question 1 is given plenty of solid background. However, Question 2 has little.**
While revising the manuscript, we changed the wording of the second question and introduced two more questions. However, we added the following paragraph that is still valid to motivate question two from the previous version of the manuscript that is still included in the new version. The paragraph reads as follows:
*"The TOC albedo is determined by all individual components of the vegetated surface (i.e., leaves, stems, soil and water content) and the structure, for example leaf clumping, of the canopy (Jones and Vaughan, 2010). Most important is the leaf area index (LAI, Watson, 1947; Asner, 1998; Jones and Vaughan, 2010), which can range from 0 to 12. It provides a measure of the total one-sided leaf area per unit of ground area, given in units of $m^2 \, m^{-2}$. The LAI itself depends on the vegetation type, follows an annual cycle, and is modulated*

*by climate conditions (Eugster et al., 2000; Davidson and Wang, 2004, 2005). A lower LAI is typically associated with an albedo that approaches the albedo of the soil, whereas an increase in LAI yields an albedo driven by the properties of the canopy (Jones and Vaughan, 2010). The second most important factor controlling canopy radiation characteristics is the leaf angle distribution (LAD, Baldocchi et al., 2002; Jones and Vaughan, 2010; Verrelst et al., 2015; Yang et al., 2023) in combination with the solar zenith angle. The LAD relates the leaf normal and the direction of the incoming radiation, which provides a measure of the sunlit leaf area with respect to the total leaf area. It is therefore a quantitative measure to describe the interaction of incoming radiation with the canopy (Asner, 1998; Stuckens et al., 2009; Vicari et al., 2019). Steeper leaf angles, such as those parameterized by an erectophile LAD, result in lower reflectivity and vice versa (Ollinger, 2011). Therefore, the combination of LAI and LAD determines how the incoming radiation will interact with the canopy and are therefore important parameters that control the CVRI."*

**Section 3.3.5: "The effect is quantified by the solar radiative forcing ΔF at the canopy level between simulations with a fixed cloud-free albedo and an albedo that accounts for cloud–vegetation-radiation interactions". I'm not convinced this is the right way to calculate this effect. Essentially, the difference here is (probably) just showing the huge difference in incoming solar radiation properties (mainly the diffuse ratio) between a cloudy and cloud-free atmosphere. This does not strictly isolate the cloud-vegetation-radiation interactions. Instead, it shows the direct effect of clouds. That's why the ΔF values are so large. You may want to do something similar to the simulations shown in Figure 2, where you run coupled and uncoupled simulations with and without clouds (over your range of optical thicknesses in Figure 9). Or reframe the section a little to clarify exactly what's being quantified and shown.**
**- Related to this comment is your conclusion section (L457-463). I don't agree with the statement on L461, as I don't believe your comparing coupled and uncoupled simulations, instead your comparing cloudy and clear-sky conditions (both in coupled mode).**

Following the Reviewer's comments, the contributions from cloud--vegetation-radiation interactions and the influence of the diffuse and direct radiation on the surface albedo are now presented separately. To avoid potential misunderstandings, the definitions of the individual radiative forcing components are explicitly given in equations 10 to 12 in the new manuscript and it is clearly stated when uncoupled and coupled models are used.
We hope this makes it clearer that we are not comparing cloudy and cloud-free simulations, as the Reviewer assumed. Due to the length of the revised section, we would like to refer the Reviewer to the new version of the manuscript.

**The language used in the manuscript is clear and consistent, yet at times very technical. I suggest making some statements more or adding clarifications to help a more general audience understand what you have done and the significance of it e.g. what does this result mean for understand land-surface radiative budgets or land-atmosphere coupling (e.g. carbon, water, energy), both now and into the future where certain properties of the land surface and atmosphere are expected to change. This is especially important when you're trying to convey a key message of the results – like in the parts of the results/discussion, conclusions and abstract. I think overall the paper would benefit significantly from this, making it much more impactful to a wider audience outside the radiative transfer modeling community.**

To better clarify the potential implications, a paragraph was added to the new subsection "Effect of neglected cloud--vegetation-radiation interactions". The paragraph reads as follows:

*"Neglecting CVRIs and the influence of clouds on the vegetation albedo introduces biases in the surface radiation budget. As shown in Fig. 9, neglecting CVRIs underestimates the amount of diffuse radiation between canopy and cloud base under illumination conditions with $\theta_0 < 60°$. This is explained by the relatively small extinction coefficient for $\theta_0 < 60°$, allowing direct radiation to penetrate deep into the canopy and to be absorbed by the canopy or soil, where it is converted into latent and sensible heat. Both fluxes are known to be important for boundary layer processes and local cloud formation (Freedman et al., 2001; Bosman et al., 2019). In contrast, a generally higher $f_{dir}$ below clouds and CVRIs increase the probability that radiation is reflected in the upper parts of the canopy and lowering the probability of absorption by the understory or soil. A second potential consequence of neglecting CVRIs is an incorrect estimate of the radiation available for photosynthesis. Due to the presence of clouds, the radiation reflected at the top of the canopy is again reflected at the cloud base that is available as diffuse radiation. It is known that diffuse radiation below optically thin clouds with $\tau \leq 6$ increases the amount of photosynthetically active radiation, since diffuse radiation can penetrate deeper into the canopy and all parts of the leaves can absorb radiation, not only the leaf areas illuminated by direct radiation (Freedman et al., 2001; Min and Wang, 2008; Kanniah et al., 2012; Cheng et al., 2016). In the same manner, the enhancement of diffuse radiation by CRVI potentially increases photosynthesis rates that will be underestimated otherwise. This results in a potential underestimation of plant productivity and carbon uptake (Freedman et al., 2001)."*

**I think there needs to be more discussion or quantification of the impact of your assumptions on your final results. This is generally lacking. For example, in your simulations "clouds were assumed to be homogeneous", only a single vertical profile of atmospheric conditions was used, assumptions embedded within the SCOPE2.0 model (e.g. it ignores woody elements, it is horizontally homogenous, has no clumping), assumptions of your canopy composition inputs (chlorophyll content, etc.). This is necessary before someone can decide to use your simpler parameterization in NWP or GCMs.**

We agree with the Reviewer's concerns about the assumptions.

In order to provide an uncertainty estimate for potential deviations from the default values that were used in libRadtran and SCOPE2.0, and to test the robustness of the simulations, an additional sensitivity analysis was performed. For the sensitivity study, an intermediate solar zenith angle $\theta_0$ of 45° and a cloud optical thickness $\tau$ of 4 were chosen. The varied parameters with their default values and ranges are listed in the following table. Each individual parameter was varied one at a time.

| Parameter | Default value | Variation |
|---|---|---|
| Atmospheric profile (Temperature and humidity) | Mid-latitude summer (afglms) | Mid-latitude winter (afglmw) US-standard profile (afglus) |
| Aerosol visibility | 50 km | 20 km, 80 km |
| Cloud altitude | 3-3.5 km | 2.5-3 km, 3.5-4 km |
| Cloud droplet effective radius | 10 μm | 7 μm, 13 μm |
| Chlorophyll AB content | 40.0 μg cm-2 | 30 (-25%), 50 (+25%) |
| Carotenoid content | 10.0 μg cm-2 | 7.5 (-25%), 12.5 (+25%) |
| Dry matter content | 0.012 g cm-2 | 0.009 (-25%), 0.15 (+25%) |
| leaf water equivalent layer | 0.009 cm | 0.00675 (-25%), 0.01125 (+25%) |
| leaf structure parameter | 1.4 | 1.05 (-25%), 1.75 (+25%) |
| BSM model parameter for soil brightness | 0.5 | 0.375 (-25%), 0.625 (+25%) |
| volumetric soil moisture content in the root zone | 0.25 | 0.1875 (-25%), 0.3125 (+25%) |

The sensitivity analysis showed that the atmospheric profile, primarily temperature and relative humidity, has the largest effect on the calculated broadband surface albedo. An absolute deviation in $\alpha_{BB}$ of up to 0.011 was determined, when the mid-latitude summer atmosphere profile was exchanged with the mid-latitude winter atmospheric profile. The influence of aerosols, cloud altitude and effective radius was found to be negligible.

With the focus of the presented study on stratiform clouds and their relative homogeneity, stratus clouds can be approximated by one-dimensional radiative transfer models. To be more specific and to follow a comment from Reviewer#2, we modified the title to better reflect the focus on stratiform liquid water clouds. The title reads as follows:

*"Impact of stratiform liquid water clouds on vegetation albedo quantified by coupling an atmosphere and a vegetation radiative transfer model".*

The sensitivity analysis with SCOPE2.0 focuses on parameters that influence the solar RT (350–2550 nm), namely chlorophyll AB content, carotenoid content, dry matter content,

leaf water equivalent layer, leaf structure parameter, BSM model parameter for soil brightness, and the volumetric soil moisture content in the root zone. Chlorophyll AB content, carotenoid content, dry matter content, leaf water equivalent layer, leaf structure parameter, BSM model parameter for soil brightness, and volumetric soil moisture content in the root zone were varied by ±25% with respect to the default value.

While a variation of ±25% may not cover the entire parameter range nor the natural variability, the sensitivity analysis is intended as a first-order approximation to estimate the impact of deviation from the default values. Because the parameters were varied one at a time, potential correlations are not resolved, for example, the interaction between the BSM model parameter for soil brightness and the volumetric soil moisture content in the root zone. Since the identified effects of parameter variation on the broadband albedo were found to be about one order of magnitude smaller compared to the effects of changing solar zenith angle and cloud optical thickness, we argue that the results of the study can be considered as a stable first-order approximation of cloud-vegetation-radiation interactions.

The results of the sensitivity study were added in form of a new subsection "Limitations of the idealized simulations" in the manuscript, where the limitations and uncertainties of the simulations are mentioned. A more detailed description of the sensitivity study is given in the new Appendix Section "Uncertainty estimates due to selection of default parameters". In addition, the caveats are made clear in Section "Summary and conclusion".

**Minor Comments**

**Can you change Equation 4 to integrate between 0.4 to 2.4 um? I can see that you clarify this in the text, but I think the stated equation needs to be the actual one you use throughout the results, and any caveats to the formal definition can be stated in the text.**

The Reviewer is right. We changed Equation 4 and the subsequent text to first describe the general case of broadband $\alpha$ and then introduce $\alpha_{BB}$ that is limited to the wavelength range from 400 to 2400 nm. The text now reads as follows:

"*Calculating the ratio between F ↑(λ) and F ↓(λ) yields the spectral albedo α(λ) (unitless) given by:*

$$\alpha(\lambda) = \frac{F^{\uparrow}(\lambda)}{F^{\downarrow}(\lambda)}.$$

*The spectrally integrated albedo α (unitless) is obtained by weighting the spectral albedo with F↓(λ) by:*

$$\alpha = \frac{\int_{\lambda_1}^{\lambda_2} \alpha(\lambda) \cdot F^{\downarrow}(\lambda) \, \mathrm{d}\lambda}{\int_{\lambda_1}^{\lambda_2} F^{\downarrow}(\lambda) \, \mathrm{d}\lambda}$$

*and integrating over the wavelength range from λ₁ to λ₂. To obtain the broadband, solar albedo, α(λ) is integrated from λ₁ = 0.2 to λ₂ = 4.5 μm, which is equivalent to measurements with broadband albedometers, i.e., a set of upward and downward looking pyranometers. In the following, the integration of α(λ) is limited to the wavelength range from 0.4 to 2.4 μm because of model constraints and indicated with $α_{BB}$.* "

**I presume you switched off the vegetation chlorophyll fluorescence calculations in SCOPE2.0? There is additional computational requirement when including fluorescence. Can you clarify in the methods whether this was on or off?**
The simulations have been performed with chlorophyll fluorescence switched on. We added a sentence in Sec. "Vegetation radiative transfer model SCOPE2.0"
*"Simulation of vegetation chlorophyll fluorescence was activated in the simulations."*

**In the text of section 2.2.2, the soil brightness parameter B = 2. However, in Table 2, B=0.5. Can you clarify which was used?**
Thank you for pointing out this inconsistency. We verified with the code and can confirm that a value of B=0.5 was used for all simulations. The typo in Sec 2.2.2. has been corrected.

**SCOPE2.0 also requires forcing inputs of meteorology e.g. temperature, humidity, wind speed, etc. What inputs did you use for the SCOPE simulations?**
As mentioned in the text, the simulations were performed with the default values of SCOPE2.0. To answer the question of the Reviewer, we provide the configuration details from section "METEO" within the SCOPE2.0 namelist file "input_data.csv": measurement height of meteorological data *z=30 m, the default value is 10 m* and we replaced it with 30 m representing the canopy height; air temperature $T_a=20°C$; broadband incoming shortwave radiation (0.4-2.5 μm) $R_{in}=600\ W\ m^{-2}$, which is overwritten by providing the spectral data; broadband incoming longwave radiation (2.5-50 um) $R_{li}=300\ W\ m^{-2}$ (incoming longwave radiation); air pressure *p=970*; atmospheric vapor pressure $e_a=15\ hPa$; wind speed at height z $u=2\ m\ s^{-1}$; atmospheric CO2 concentration $C_a=410\ ppm$; atmospheric $O_2$ concentration $O_a=209\ ppm$. Since the focus of the study is on simulations in the solar wavelength range, meteorological parameters like temperature or wind are of minor importance. However, we acknowledge that these parameters will become important, when considering the thermal-infrared wavelength range.

**In Figure 2a and 2b, do the colors represent the same simulations? From what I can understand, they are completely different simulations e.g. the red line in 2a is "uncoupled, including clouds", but in 2b the red line is "coupled, neglecting clouds". I think you need to use different colors to avoid confusion here.**
The Reviewer is right. The colors represent different simulations. We followed the

suggestion and now use different colors to better separate the different simulations.

**I suggest reducing the y-axis limits in Figure 3g and 3h (perhaps from 0.5 to 1.5), to make the differences more clear.**
We do agree with the Reviewer. We new use a range between 0.7 and 1.8 for the *y*-axis.

**Should the black line (tau=0) also be plotted in Figures 3a, 3b, 3g, 3h, Figure 4c, and 4d? I know it will only be a straight line = 1, but I think it is still important to plot it for consistency and clarity.**
We followed the suggestion of the Reviewer and added the black line representing a cloud optical thickness of τ=0.

**Can you clarify in the captions of Figure 3 and Figure 4 the LAD used. That needs to be highlighted more clearly as the difference between these two figures. You could even include text annotation inside the subplot to make it more clear to the reader.**
We agree with the Reviewer and have added titles to both plots. In addition, the figure captions now indicate which leaf angle distribution is used.

**Please go through and re-check the spelling. There are many instances where words are misspelled or grammar needs revising (e.g. L57, L367, Section 3.3.5 title, L423, L455).**
After creating a new version of the manuscript, we checked for typos and grammar.

**Section 3.3.6: Can you provide some more detail on why a simple parameterization is useful or necessary? Given the complexities and non-linearity of these relationships in your results, one could argue that use of these coupled, spectrally-resolved RT models is necessary for a more complete and robust representation of the land surface radiative budget and albedo effects.**
The Reviewer raises an important point.
We added the following sentences to the introduction:
*"Land-surface–atmosphere interactions are a key concern in dynamic modeling (Ardaneh et al., 2025). In the context of radiative processes, the spectral surface albedo α(λ), with λ the wavelength, determines the extent to which solar radiation is absorbed and reflected by the Earth's surface. The surface albedo determines the surplus of energy that is transferred into sensible and latent heat (Moene and Van Dam, 2014). Consequently, the integrated surface albedo α is a central factor in numerical weather prediction models and global climate models. Both types of models simulate the interaction between the atmosphere and the surface, and a realistic representation is crucial since cloud–*

*vegetation interactions via surface flues, for example, can reinforce the thickness of shallow stratocumulus (Freedman et al., 2001; Zhang and Klein, 2013). However, the implementation of vegetation albedo in numerical weather prediction and global climate models is often simplified by using climatologies but neglecting cloud–vegetation–radiation interactions (CVRIs) for example in the Integrated Forecasting System (IFS) of the European Centre for Medium-Range Weather Forecasts (ECMWF) (ECMWF, 2021)."*
Later, this is point is picked up again at the beginning of subsection "Parameterization of cloud-vegetation-radiation effects". It begins with a short paragraph discussing the current parametrization of surface albedo in models. This helps in understanding the use of the proposed "simple" parametrization:
*"Within ECMWF IFS, the vegetation albedo is based on monthly climatologies of LAI and vegetation type, and thus indirectly on the LAD (ECMWF, 2021). No separation between black-sky and blue-sky albedo is made (ECMWF, 2021). Consequently, neither the influence of clouds on vegetation albedo nor CVRIs are considered in RT simulations performed within ECMWF IFS or models with similar albedo implementation."*

**Section 3.3.6: Your equation 11 has fdir as being wavelength dependent, yet the calculated broadband albedo is not. Should the fdir input on the right-hand side actually be the average broadband fdir? Or should the broadband albedo actually be spectral albedo?**
The Reviewer is correct and addresses an important point.
We revised Eq.2 and included a definition for the broadband direct fraction, which is Eq.3 in the new manuscript.
The former Eq.11, which is now Eq.13, is also changed to included the broadband direct component. The text in section "Parameterization of the cloud effect on broadband surface albedo" has been changed accordingly. Since modifications have been made in several instances, we would like to ask the Reviewer to see the changes in the track-changes manuscript.

**Section 3.3.6: The other caveat of this parameterization is that it is limited to the idealized conditions of your simulations. So, it is subject to your other assumptions e.g. homogenous cloud cover, assumed vertical profiles in the atmosphere (aerosols, temp., humidity, gas conc.), assumptions in SCOPE2.0 (no woody elements or canopy clumping considered, fixed soil background and moisture conditions), etc. In addition, it requires LAI to be greater than 2, which is not the case for large portions of the land surface and many times in the seasonal cycle of both deciduous forests and grasslands/croplands.**
We do agree with the Reviewer that excluding LAI smaller than 2 is a caveat. Therefore, we extended the simulated range to include LAI = 1 and the parametrization. The polynomial fits in Table 3 and Figure 8 (in the new version) have been updated.
The impact of idealized simulations was addressed by performing a sensitivity study. Please see the reply to mayor comment #5.

**- These caveats must also be made clear in the conclusions (L464-471).**
The caveats of the idealized simulations and potential uncertainties are now also

mentioned in Section "Summary and conclusion":
*"The idealized simulation set-up and the multitude of vegetation parameters did not allow to cover the natural variability of atmospheric and vegetation conditions, and the chosen default values may affect the results presented in this study. A sensitivity study was performed to estimate the influence of these default parameters and to test the robustness of the results. Varying the input parameters of the atmospheric RT model showed that the largest uncertainties in $\alpha_{BB}$ of up to 3.8% result from the variation of the atmospheric profile. Variations in aerosol visibility, cloud altitude, and cloud droplet effective radius contributed only little. Among the varied parameters, plant dry matter had the largest effect on $\alpha_{BB}$, followed by the assumed atmospheric profile and the leaf structure parameter. Varying the default values by ±25 % caused deviations in $\alpha_{BB}$ of up to ±0.013, which corresponds to a change in LAI of about 1. Variations in chlorophyll AB content, carotenoid content, leaf water equivalent layer, BSM model parameter for soil brightness, and the volumetric soil moisture content in the root zone had only small effects. It is also acknowledged that the simulations assume a homogeneous canopy and structural effects such as leaf clumping were not considered in this study. However, these structural effects operate on a local scale and are likely to be smoothed out given the current spatial resolution of numerical weather prediction models and global circulation models"*

**L442-443: "The LAI was found to have the largest impact on the resulting spectral and broadband α". I disagree with this statement. The sensitivities you have evaluated across different inputs are not necessarily comparable, as the various inputs have different units, meanings, and ranges over which they are evaluated. More importantly, I can see from Fig. 6 a, d, g that for a zenith angle of 25° and optical thickness < 2, the change in LAD has a much larger effect on albedo (a change of up to 0.09) than the change in LAI from 2 to 5 (a change of less than 0.02). I think this statement needs revising and perhaps reconsider how you quantify which inputs have the "largest impact" i.e. show most sensitivity.**

We do agree with the Reviewer and modified this section, which now reads as follows:
*"Based on the varied parameters and parameter ranges, it was found that the LAD is the primary factor controlling the sensitivity of $\alpha_{BB}$ to LAI, $\theta_0$, and $\tau$ . Assuming an erectophile LAD in the simulations, $\alpha_{BB}$ was most sensitive to the varied parameters, especially for combinations of small $\tau < 6$ and small $\theta_0 < 50°$, i.e., large values of the direct fraction $f_{dir}(\lambda)$. Generally lower sensitivities of spectral and broadband α to $\tau$ and $\theta_0$ were found for the spherical LAD. Spectral and broadband α of the planophile LAD were found to be almost insensitive to $\tau$ and $\theta 0$ for the same parameter ranges. The sensitivity of $\alpha(\lambda)$ to LAI, LAD, and $\theta_0$ decreased continuously with decreasing fraction $f_{dir}(\lambda)$ because the incident radiation becomes more diffuse, i.e., undirected, and the angular-dependent scattering in the canopy becomes insensitive to the canopy structure given by LAI and LAD. The second effect that affected the spectrally integrated broadband albedo $\alpha_{BB}$ was the wavelength-dependent absorption and scattering by clouds, which shifted the weight of the incoming radiation towards shorter wavelengths. Due to the generally low values of $\alpha(\lambda)$ below 700 nm, the effect of the wavelength shift was found to be small in absolute values, increasing $\alpha_{BB}$ by up to 0.07 ($\theta_0 = 70°$ and $\tau = 6$). In summary, the change in $f_{dir}(\lambda)$ is*

*relevant for values of τ between 0 and 6, when diffuse radiation is dominant. Beyond τ of 6, the shift in the spectral weighting of α(λ) with F ↓(λ) is still relevant."*

**L449-450: "This is caused by the dominating fraction of isotropically reflected radiation from the surface that is less sensitive on the incident angle of the radiation compared to the reflection of direct radiation." Can you add a clarification for a more general audience? I believe a more simple statement is that "as the incoming radiation becomes more diffuse, the effects canopy structure (LAI, LAD) and solar zenith angle on surface albedo become minimal".**

We do agree with the Reviewer and rephrased the sentences as follows:

*"The sensitivity of α(λ) on LAI, LAD, and $\theta_0$ decreased continuously with decreasing fraction fdir(λ) since the incoming radiation becomes more diffuse, i.e., undirected, and the angular dependent scattering in the canopy becomes insensitive to the canopy structure, given by LAI and LAD, and $\theta_0$."*

**L9-11: "The greatest albedo increase is observed during the transition from cloud-free to cloud conditions with a cloud optical thickness (τ) of about 6". This statement could be misinterpreted as the greatest sensitivity occurs when τ is around 6. I think what you really mean is that the greatest sensitivity occurs in the range from cloud-free (τ =0) to cloud optical thicknesses of about 6.**

The Reviewer is right. We changed the sentence as suggested:

*"The greatest increase in albedo is observed during the transition from cloud-free to cloudy conditions with a cloud optical thickness (τ ) in the range between 0 and 6"*

---

## Author Comment (AC2)

**Reply to Reviewer #2**

(Referee comment on "Impact of clouds on vegetation albedo quantified by coupling an atmosphere and a vegetation radiative transfer model" by K. Wolf et al. (egusphere-2024-3614),https://doi.org/10.5194/egusphere-2024-3614-RC2, 2025)

We thank the Reviewer for the time she/he spent on the manuscript. We appreciate the provided comments and we believe that the manuscript has been significantly improved by incorporating the Reviewer's suggestions.

To improve readability and logical structure, the order of sections and subsections has been revised. We hope that this will allow the reader to better navigate through the different points discussed in the manuscript. The manuscript has also been extended to include a discussion of modeling limitations and the influence of the selected default values.

For better legibility, the Reviewer's comments are highlighted in **bold** and changes in the manuscript are in *italic*.
* * *
**This is a modelling study applying coupled RTM in the atmosphere and vegetation canopy to quantify vegetation albedo under cloudy conditions. The study is timely as understanding of biophysical forest effects on radiation balance and further on climate are essential for forest management strategies. However, the structure and content of the manuscript has to be significantly improved before it can be published. In addition, there are a lot of small misprints in the text, and I encourage the authors to read the manuscript thoroughly during the next iteration.**

**Major comments**

1. **The authors put a lot of efforts in the description of the models, which is done very good, while all other parts got less attention. Results section is currently a mixture of methods and results. I would strongly recommend describing all the modelling setups and their purposes in the Methods section, making a separate subsection, where this information can be added.**
   We respectfully disagree with the Reviewer's comment regarding the mixing of methods and results. If the Reviewer is referring to Eq.9 (old version of the manuscript), the definition of the relative contribution of multiple scattering, and Eq.10 (old version of the manuscript), the definition of the solar albedo-forcing that appear in the result section, then we think that it is appropriate to define these along the way in the results section. The definitions themselves are short and best explained in the context of the surrounding text. Presenting the definitions out of context in a separate "Method" section would make them appear as a surprise.

   **The authors could also explain better why they consider this range of variables or specific variables they chose for simulations. Otherwise, all the results come as a surprise.**

It is not exactly clear to us what the Reviewer means by "this range of variables or specific variables" since it is not specified to which part of the manuscript the Reviewer is referring.

In our understanding, the choice of parameters for the atmospheric model is explained in Section "Atmospheric radiative transfer model libRadtran", where the model set-up is described, parameter ranges are given, and supported by the cited literature.

For the vegetation part, the simulations mostly follow the default parameter settings of SCOPE2.0, which are considered to be a reasonable first guess for generic vegetation / a generic canopy. The spherical leaf angle distribution (default of SCOPE2.0) was chosen as the general case, and the planophile and erectophile leaf angle distributions were chosen as extremes. The leaf area index with a value of 3 (default in SCOPE2.0) was initially varied from 2 to 5 to obtain a range / variety around the default. The LAI was extended to include LAI=1.

The chosen values of cloud optical thickness $\tau$ from 0 to 4 and 6 in Figures 3, 4, 6, and 7 were chosen because this is the range where the direct fraction of downward irradiance is most sensitive to changes in $\tau$. This is also explained in the text.

For Figure 6 the $x$-axis was extended to include $\tau = 60$ since a different process, here multiple scattering, was investigated.

We rephrased the following lines in the manuscript to be more specific:

*"The solar zenith angle $\theta_0$ was varied between 25° and 70° to cover the typical range of the mid-latitudes."*

*"The cloud optical thickness $\tau$ , was varied between values of 0 and 80 to simulate natural conditions of stratus clouds ranging from cloud-free to densely overcast, and to include values of $\tau$ at which the cloud top reflectivity saturates (King, 1987; Nakajima et al., 1991; Tselioudis et al., 1992). "*

*"Analyzing the coupled simulations, it was found that the sensitivity of the simulated spectral and broadband $F(\lambda)$ and $\alpha(\lambda)$ was greatest below $\tau = 6$, thus, defining the range of $\tau$, which is most interesting for understanding cloud–vegetation–radiation interactions. The simulations are performed for an intermediate $\theta_0 = 45°$ and $\tau = 2$, where diffuse radiation dominates but direct radiation is still contributing to the radiation field."*

**In addition, this manuscript lacks discussion part. How do all these results compare to the previous findings? At least some of these effects were previously reported in other studies, for example, the change in albedo between low-diffuse- and high-diffuse-fraction conditions at low and high zenith angles.**

Following the Reviewer's comment we have placed  our results in the context of existing literature. The following sentences and paragraphs have been expanded and added to the manuscript.

*"Cloud–vegetation-radiation interactions, here primarily multiple scattering between cloud base and the canopy, are known to enhance the observed spectral albedo (Weihs et al., 2001; Wendisch et al., 2004; Gueymard, 2017). The enhancement is caused by an additional contribution of radiation to $F^{\downarrow}_{dif}(\lambda)$ that was reflected at the TOC back to the atmosphere and again back to the canopy by the cloud base*

*(Freedman et al., 2001; Min and Wang, 2008; Kanniah et al., 2012; Gueymard, 2017). ..."*

*"Generally, all cases with high surface albedo, i.e., over snow and ice covered areas, are prone to enhanced diffuse radiation and albedo below clouds (Hay, 1976; Kierkus and Colborne, 1989; Gueymard and Ruiz-Arias, 2016; Gueymard, 2017). However, the absolute enhancement of diffuse radiation above vegetation is smaller due to the generally lower albedo compared to ice."*

*"Although vegetated surfaces have a lower spectral albedo compared to Arctic regions, it can be expected that clouds have a similar effect on the TOC albedo. For example, Gueymard (2017) showed that clouds can enhance the broadband albedo, also called albedo enhancement, by backscattering radiation at cloud base towards to surface, which leads to an increase in the diffuse downward irradiance. Neglecting potential albedo enhancements in models may cause biases in the simulated radiative budget. Furthermore, it is known that very thin cloud layers with $\tau \leq 6$ tend to increase the diffuse downward irradiance that can penetrate deeper into the canopy and enhance the photosynthesis rate, which is also called the diffuse fertilization effect. A further increase in $\tau$ then leads to an overall reduction in downward irradiance and lower photosynthesis rates (Freedman et al., 2001; Min and Wang, 2008; Kanniah et al., 2012). "*

*"For values of $\theta_0$ of 25° and 50°, $\alpha_{BB}$ is lowest for the black-sky albedo, while the highest values of $\alpha_{BB}$ are found for the white-sky albedo. The black-sky and white-sky albedo increase with increasing $\tau$ . The blue-sky albedo, as an intermediate condition between the black-sky and white-sky albedo, is closest to the black-sky albedo for cloud-free conditions and approaches the white-sky albedo under overcast conditions ($\tau > 6$). This agrees with the observations by Freedman et al. (2001) and Kanniah et al. (2012), who found an increase in the canopy albedo under cloudy conditions compared to clear-sky conditions."*

**2. The title and abstract refer to clouds in general but in fact, as far as I could see, simulated clouds represent liquid altostratus, which I think should be mentioned explicitly already in the introduction, and a couple of words can be said about justification of this choice.**
We agree with the Reviewer that we need to be more precise in this regard. The title has been rephrased as follows:
*"Impact of stratiform liquid water clouds on vegetation albedo quantified by coupling an atmosphere and a vegetation radiative transfer model".*

Stratiform liquid water clouds have been selected since this is a common cloud type over land. In addition, one-dimensional radiative transfer simulations best resemble stratiform clouds.
*"Since one-dimensional RT models assume homogeneous clouds that best resemble stratiform clouds, this study focuses on low- and mid-level warm stratus and altostratus, which cover between 4 and 12 % of the entire globe at any time (Eastman et al., 2011). These clouds are represented by liquid water clouds with a*

*fixed cloud base at an altitude of 3 km and a cloud top altitude of 3.5 km representative of mid-latitude, continental clouds. The cloud droplet effective radius is fixed to 10 μm that is typically found in such clouds over continents (Stephens, 1994; Frisch et al., 2002; Aebi et al., 2020). "*

3. **Related to the estimates of the radiative forcing:**
   **first, it is not mentioned at all anywhere before the corresponding Results subsection starts;**
   The following sentence was modified and added in the introduction:
   *"For example, Gueymard (2017) showed that clouds can enhance the broadband albedo, also called albedo enhancement, by backscattering radiation at cloud base towards to surface, which leads to an increase in the diffuse downward irradiance. Neglecting potential albedo enhancements in models may cause biases in the simulated radiative budget"*
   *(Please also see the answer to major comment #1.)*

   **second, I do not really understand why it is done for clear sky downwelling irradiance if the whole point of the study is that albedo is calculated for cloudy conditions, and changes are associated with the present clouds. To me, it would make more sense either to compare the difference in radiation balance between cloudy coupled and uncoupled simulations or cloudy-coupled vs clear-sky albedo for cloudy conditions.**
   Following the Reviewer's comments the contributions from cloud--vegetation-radiation interactions and the influence of the diffuse and direct radiation on the surface albedo are now presented separately. To avoid potential misunderstandings, the definitions of the individual radiative forcing components are explicitly given in equations 10 to 12 in the new manuscript and it is clearly stated when uncoupled and coupled models are used.
   We hope this makes it clearer that we are not comparing cloudy and clear-sky simulations, as the Reviewer assumed. Due to the length of the revised section, we would like to refer the Reviewer to the new version of the manuscript.

4. **Results section starts with a long text, which in fact represents a separate subsection. Please make it a subsection and give it a title. Instead, a reader would appreciate a brief summary of the story in the different Results subsections right after the title Results and Discussion. Oppositely, subsection 3.3.1 is too short, and it is not clear to me why it is separated from the next subsection which discusses panels of the same figure. Related to that, Subsection 3.3 is called simply 'Broadband' and must be renamed.**
   We do agree with the statement of the Reviewer and completely revised the structure of the paper in order to facilitate the readability. Due to the comprehensive re-ordering we would like to direct the Reviewer to the new version of the manuscript.

**Minor comments:**

**L18-19: 'an important boundary between the lithosphere and atmosphere, across which energy fluxes (latent and sensible heat, turbulence, gases, aerosol particles, and radiation) are exchanged' – please rewrite, turbulence and aerosol particles are not present in the lithosphere**
The first sentence has been rephrased as follows:
*"The Earth's surface represents an important boundary between the lithosphere and atmosphere, through which energy fluxes (latent and sensible heat, turbulence, gases, aerosol particles, and radiation) are exchanged. Land-surface–atmosphere interactions are a key concern in dynamic modeling (Ardaneh et al., 2025). In the context of radiative processes, the spectral surface albedo α(λ), with λ the wavelength, determines the extent to which solar radiation is absorbed and reflected by the Earth's surface. The surface albedo determines the surplus of energy that is transferred into sensible and latent heat (Moene and Van Dam, 2014)."*

**L60-61 'As a result of this discussion, there are two question to be addressed in this paper' – please link the previous discussion and the research questions better and state explicitly the novelty of the current approach. To me, it clearly comes later in lines 136-138.**
We followed the suggestion of the Reviewer and rephrased the section as follows:
*"As a result of this discussion and since the radiation interactions of clouds and vegetation have not been explicitly simulated yet, the following four questions are addressed in this paper:*

*i How strongly do clouds impact the spectral and broadband albedo of vegetation?*

*ii How large are the improvements in broadband albedo by coupling atmospheric and vegetation RT models?*

*iii Can we separate and quantify individual coupling effects?*

*iv What are the consequences for cloud radiative effects?*

*To answer the above questions and to systematically investigate CVRIs, we iteratively coupled the atmospheric RT model libRadtran and the vegetation RT model SCOPE2.0 to investigate the radiative interaction of clouds and vegetation. The model coupling provides a more realistic input to the atmospheric radiative transfer model libRadtran by incorporating the vegetation albedo from SCOPE2.0, while the simulated spectral downward irradiance from libRadtran fed into SCOPE2.0 accounts for scattering and absorption by clouds."*

**L 197: tau = 80 is optically thick overcast. So far it looks the authors simulate mid-level clear sky and overcast mid-level liquid clouds (altostratus) with different cloud thickness.**

In section "2.2.1 Atmospheric radiative transfer model libRadtran" we better describe the selection of the cloud optical thickness values by:

*"...The cloud optical thickness τ , was varied between values of 0 and 80 to simulate conditions ranging from cloud-free to densely overcast. ..."*

If this does not adequately address the Reviewer's comment and the Reviewer would like us to pursue this comment further, we ask the Reviewer to be more specific in his or her request.

**L 202: tau = 2 mentioned but then in Fig. 2 tau is 4**

The Reviewer is right. The value of τ=4 was incorrect in the figure caption and is now corrected to τ=2.

**Fig. 2: In legends, check 'uncoupled' and dots are not needed. Related to inserts, the choice of coupled simulations albedo as a reference value is counterintuitive. I'd prefer to see the change in albedo quantified with regards to uncoupled simulations.**

The labels have been changed. Together with the modifications in the text, we hope that this better clarifies what is shown in the plots and avoids misunderstandings. The coupled simulations were chosen as the reference because they represent irradiance and albedo closer to the values that would be observed in nature. Therefore, we keep the coupled simulations as the reference.

Please note that Fig.2 has been updated to be more precise in the wording and to address the Reviewer's concern raised in the comment.

**L213: 'different stages of coupling'. Is clear-sky case really the stage of coupling for cloudy conditions? I think the authors speak of different setups here.**

To be more clear, the color code and the legend in Fig.2 have been updated. In this way, simulations from panel a and b cannot be confused. The adjoined text is also revised to better express, why we consider cloud-free simulations and cloudy simulations with and without coupling. The text now reads as follows:

*"The necessity to initialize vegetation RT simulations with realistic $F^{\downarrow}(\lambda)$ and the need for RT model coupling is demonstrated in Fig. 2a and b, which show an example of $F^{\downarrow}_{dif}(\lambda)$ and $\alpha(\lambda)$ at different stages of model coupling and iteration. The simulations are performed for an intermediate $\theta_0 = 45°$ and $\tau = 2$, where diffuse radiation dominates but direct radiation is still contributing to the radiation field. Under cloud-free conditions (black line), downward diffuse irradiance $F^{\downarrow}_{dif}(\lambda)$ above the canopy is generally small, except below 700 nm wavelengths, where the contribution from Rayleigh scattering increases. Including clouds in the atmospheric RT model increases $F^{\downarrow}_{dif}(\lambda)$ (red line) compared to the cloud-free case due to scattering at cloud particles. The spectrum is characterized by water vapor*

*absorption in the wavelength bands of 933–946 nm, 1118–1144 nm, 1350–1480 nm, and 1810–1959 nm. Coupling of libRadtran and SCOPE2.0 iteratively results in $F\!\downarrow_{dif}(\lambda)$ (orange line) which is slightly higher compared to the uncoupled simulations since multiple-scattering enhances $F\!\downarrow_{dif}(\lambda)$. Relative differences, given in the subpanel of Fig. 2a, up to −5% are identified between the uncoupled and coupled cloudy simulations for wavelengths between 700 and 1200 nm, where the total $F\!\downarrow$ and $\alpha(\lambda)$ are generally largest. Since clouds modify $F\!\downarrow(\lambda)$ spectrally and $f_{dir}(\lambda)$, they also impact $\alpha(\lambda)$. Figure 2b shows $\alpha(\lambda)$ during different model set-ups and iterations. A generic $\alpha(\lambda)$ is provided by the IGBP data base (brown line), which was also used as a first-guess to initialize the libRadtran simulations. The radiation is reflected isotropically and does not take into account any dependence on the incident angle nor the presence of clouds. Running SCOPE2.0 freely without any constraints from the atmosphere, i.e., assuming a cloud-free atmosphere, a better resolved $\alpha(\lambda)$ is obtained (gray line). By providing spectra of direct and diffuse $F\!\downarrow(\lambda)$ that represent cloudy conditions with $\tau = 2$, a higher $\alpha(\lambda)$ is obtained (blue line), which is caused by the greater fraction of diffuse radiation. Relative differences of about −2 to 15 % are determined. Simulations at this stage of the iteration still neglect CVRIs. Coupling both models under cloud conditions results in $\alpha(\lambda)$ (green line), which is slightly further enhanced compared to the blue line due to multi-scattering between TOC and cloud base. For the given example, the relative differences range between 3 and −5 %, with respect to the fully coupled simulations (see subpanel Fig. 2b). The following analysis will systematically examine the discrepancies in spectral and broadband $F(\lambda)$ and $\alpha$ between uncoupled and coupled simulations, depending on $\theta_0$ and the optical properties of clouds and vegetation."*

**Fig. 3 panel b: why are there increases of this ratio above 1 close to absorbing intervals at large theta? Are these some artifacts resulting from too low radiation in the denominator?**
The Reviewer is correct that these are due to artifacts in the denominator. The numerically unstable values have been masked and an updated version of the figure is now included in the manuscript.

**Add in the captions that Fig. 3 uses spherical LAD and Fig. 4 erectofile**
The Reviewer is right. We added titles to Fig. 3 and 4. In addition, we explicitly mention the LAD in the figure captions:
*"Simulations for a solar zenith angle $\theta_0 = 25°$ (left column) and $\theta_0 = 70°$ (right column), a leaf area index LAI = 3, and a spherical leaf angle distribution. "*
*"Simulations for a solar zenith angle $\theta_0 = 25°$ (left column) and $\theta_0 = 70°$ (right column), a leaf area index LAI = 3, and an erectophile leaf angle distribution"*

**Fig. 6: I'd prefer to see legends in these figures than overlapping numbers**

Due to the number of different colors and the fact that there are two parameters, solar zenith angle and cloud optical thickness, we still think that writing the values next to the lines is easier to interpret. However, we do agree with the Reviewer and rearranged the labels to avoid the overlap.

**L 359: Fig. 7(d-f) - no such figures**

The Reviewer is correct. The figure has been modified during the preparation of the manuscript; some panels were removed, but the reference was forgotten in the text, which is now removed.